# Determination of equivalent black carbon mass concentration from aerosol light absorption using variable mass absorption cross-section

Weilun Zhao[1], Wangshu Tan[1,2], Gang Zhao[1,3], Chuanyang Shen[1], Yingli Yu[4,1], Chunsheng Zhao[1]

[1]Department of Atmospheric and Oceanic Sciences, School of Physics, Peking University, Beijing 100871, China

[2]School of Optics and Photonics, Beijing Institute of Technology, Beijing 100081, China

[3]State Key Joint Laboratory of Environmental Simulation and Pollution Control, College of Environmental Sciences and Engineering, Peking University, Beijing 100871, China

[4]Economics & Technology Research Institute, China National Petroleum Corporation, Beijing 100724, China

*Correspondence to*: Chunsheng Zhao (zcs@pku.edu.cn)

**Abstract.** Atmospheric black carbon (BC) is the strongest solar radiative absorber in the atmosphere, exerting significant influences on the earth's radiation budget. The mass absorption cross-section (MAC) is a crucial parameter for converting light absorption coefficient ($\sigma_{ab}$) to equivalent BC mass concentration (EBC). Traditional filter-based instrument, such as AE33, uses a constant MAC of 7.77 $m^2/g$ at 880 nm to derive EBC, which may lead to uncertainty in EBC. In this paper, a new method of converting $\sigma_{ab}$ to EBC is proposed by incorporating the variations of MAC attributed to the influences of aerosol coating state. Mie simulation showed that MAC varied dramatically with different core sizes and shell thicknesses. We compared our new method with traditional method during a field measurement at a site of North China Plain. The results showed that the MAC at 880 nm was smaller (larger) than 7.77 $m^2/g$ for particle smaller (larger) than 280 nm, resulting in EBC mass size distribution derived from new method was higher (lower) than traditional method for particle smaller (larger) than 280 nm. Size-integrated EBC derived from the new method was 16% higher than traditional method. Sensitivity analysis indicated that the uncertainty in EBC caused by refractive index (RI) was within 35% and the imaginary part of RI had dominant influence on the derived EBC. This study emphasizes the necessity to take variations of MAC into account when deriving EBC from $\sigma_{ab}$ and can help constrain the uncertainty in EBC measurements.

## 1 Introduction

Black carbon (BC) is an important component of atmospheric aerosol particles. The warming effect of BC is only second to that of carbon dioxide (Ramanathan and Carmichael, 2008) because of its highly absorbing property. The environmental effect of BC is nonnegligible. The absorption of BC can significantly reduce visibility (Moosmuller et al., 2009). BC are considered a major factor of adverse health disease (Highwood and Kinnersley, 2006). The fractal aggregates morphology of BC provides substantial surface area for deposition of cancerogenic matter. The insoluble nature and fine size of BC make it deposit in the lung for a long time. Because the significant impact of BC, extensive measurement has been made to monitor atmospheric loading of BC and give reference to policymaker for mitigation.

The BC mass concentration ($m_{BC}$) is one of the important variables for BC measurement (Bond et al., 2013). Many methods have been proposed to determine $m_{BC}$. For instance, the single-particle soot photometer (SP2) measure refractory BC (rBC) based on laser-induced incandescence (Schwarz et al., 2006). The organic carbon/elemental carbon (OCEC) analyzer determines elemental

carbon (EC) through heating collected sample in a subsequent helium/oxygen environment (Wu et al., 2012). Soot particle aerosol mass spectrometry (SP-AMS) combines laser-induced incandescence as well as laser vaporization used in mass spectrometry (Onasch et al., 2012) and also reports $m_{BC}$ as rBC. However, the abovementioned instruments are complicated in structure, highly expensive, hard to maintain, as a result, not widely used.

Filter-based instruments, such as aethalometer (Hansen et al., 1984), are commonly used for routine BC observations and dedicated campaigns (Castagna et al., 2019;Sandradewi et al., 2008;Helin et al., 2018) because they are convenient and easy to maintain. Aethalometer does not directly measure $m_{BC}$ and actually measures light absorption. Aethalometer converts absorption coefficient ($\sigma_{ab}$) at 880 nm to equivalent BC mass concentration (EBC) (Petzold et al., 2013) through a fixed mass absorption cross-section (MAC, 7.77 $m^2/g$ at 880 nm). However, field measurements indicated that MAC showed both large temporal and spatial variability (Bond and Bergstrom, 2006;Lack et al., 2012;Cappa et al., 2012). For example, Bond et al. (2006) reported MAC at 550 nm varying from 1.6 to 15.9 $m^2/g$. Sharma et al. (2002) reported MAC at 880 nm varying from 6.4 to 28.3 $m^2/g$. It is not appropriate to use a fixed MAC at when EBC is derived from $\sigma_{ab}$ at 880 nm. The variation of MAC has to be taken into account to reduce the uncertainty in the $\sigma_{ab}$-derived EBC.

The mixing state of BC is one of the crucial reasons leading to large variation in MAC. Field measurements have indicated that fresh BC particles are generally subject to several coating processes while being transported in the atmosphere and tend to be covered in layers of other organic or inorganic components (Shiraiwa et al., 2007;Cappa et al., 2019;Bond et al., 2006). The gathered shell that builds up on the BC core, acting as a lens to focus additional incident light on the enclosed BC core, can enhance BC light absorption (Fuller et al., 1999). As a result, a coated BC particle will have a bigger MAC than the original pure BC particle. This light absorption enhancement is termed as "lensing effect" of the BC-containing particles. For typical core-coating mixed BC-containing particles, this lensing effect was found to enhance BC absorption by 50-100% (Bond et al., 2006). Schwarz et al. (2008) found that fresh soot particles internally mixed with sulfates and organics during transportation, and the lensing effect enhanced the light absorption by a factor of 1.3-1.5.

At a given wavelength, such as 880 nm, the value of MAC relies on the size and the location of BC core, coating thickness, as well as refractive index (RI) (Fuller et al., 1999;Lack and Cappa, 2010). A simplified core-shell configuration has been introduced to illustrate the structure of BC-containing particles and calculate the relevant optical properties. Several studies have demonstrated that it is appropriate to use the core-shell configuration for aged aerosol (Majdi et al., 2020;Liu et al., 2019;Li et al., 2019).

In the previous studies (Zhao et al., 2019b;Ran et al., 2016a;Ran et al., 2016b;Castagna et al., 2019), the variation of MAC due to mixing state was not considered when deriving EBC from $\sigma_{ab}$. With the objective of improving the reliability of $\sigma_{ab}$-derived EBC, the Mie model incorporated with core-shell configuration hypothesis was applied in this study to assess the limitation of fixed MAC on $\sigma_{ab}$-derived EBC. Based on the detailed analysis of the relationship among MAC, $D_{BC}$, and coating thickness ($T_{shell}$), a modified approach considering variation of MAC due to mixing state was proposed for filter-based instruments to derive EBC from $\sigma_{ab}$. Detailed uncertainty analysis is carried out to assess the influence of assumptions used in this study. This modified method estimates size-resolvoled EBC accurately and reduces the uncertainty in $\sigma_{ab}$-derived EBC with respect to mixing state.

## 2 Dataset and instrumentation

### 2.1 Measurement sites

The EBC particle mass size distribution (BCPMSD) was obtained at the Zhangqiu Meteorology Station (36°42'N, 117°30'E), Shandong Province, surrounded by farmland and residential areas and a typical site for regional background conditions of North China Plain (NCP). The field campaign lasted for about 1 month, from July 23, 2017 to August 24, 2017.

The number fraction of BC-containing aerosol ($N_{BC}$) is required during conversion from absorption to EBC. $N_{BC}$ was not measured simultaneously at Zhangqiu due to limitation in instruments. $N_{BC}$ is a reference value in this work and referred from measurement at Taizhou (32°35'N, 119°57'E). An SP2 was used to determine $N_{BC}$ at Taizhou from May 24, 2018 to June 18, 2018. The suburban site Taizhou lies at the south end of Jianghuai Plain in the East of China. This industrial area between the two megacities of Nanjing and Shanghai has experienced severe pollution during the past thirty years. Hence, $N_{BC}$ measured at Taizhou is representative and the campaign averaged $N_{BC}$ is used in this work. Besides, comparison between AE33 and the three-wavelength photoacoustic soot spectrometer (PASS-3) at 405 nm, 532 nm and 781 nm (Zhao et al., 2020) was also carried out at Taizhou for scattering correction of AE33.

Besides Taizhou, the comparison between AE33 and PASS-3 was also conducted from March 20, 2018 to April 30, 2018 and from October 10, 2018 to October 19, 2018 in Peking University (39°59'N, 116°18'E). This site is located at the northwest of Beijing, a megacity experiencing severe and complex urban pollution. From March 21, 2017 to April 9, 2017 at the Peking University site, simultaneous measurements of aethalometer AE51 (model 51, microAeth, USA) and AE33 at 880 nm were carried out to investigate the consistency between AE51 and AE33.

### 2.2 Instruments

All the measurements in the three sites were conducted in temperature ($24 \pm 2$ °C) controlled containers, and a particle pre-impactor was used to remove particles larger than 10 μm from the input air stream. The drying systems in the three sites were configured with a Nafion dryer to keep the relative humidity of sample flow below 40%. This type of dryer performs well in reducing aerosol losses. The transmission efficiency of the Nafion dryer is up to 90% for particles smaller than 10 nm and rises up to 100% for particles larger than 30 nm (The performance details of the Nafion dryer can be accessed at http://www.permapure.com).

During the field campaign at the Zhangqiu site, the particle number size distribution (PNSD) as well as BCPMSD were simultaneously determined using the measurement system developed by Ning et al. (2013) and improved by Zhao et al. (2019b). The polydisperse aerosol sample flow was first drawn into DMA (Model 3080, TSI, USA) to select relatively monodispersed aerosol sub-populations with diameters ranging from 97 to 602 nm. Sheath and sample flows were set as 3 and 0.5 L/min, respectively. The selected monodispersed aerosol populations were further divided into two paths. One path (0.2 L/min) was drawn into AE51 for EBC measurements. The other path (0.3 L/min) was analyzed using CPC (model 3772, TSI, USA) for number concentration measurements. As the standard sample flow for CPC 3772 is 1 L/min, a cleaned airflow of 0.7 L/min was added for compensation. A BCPMSD cycle measured here required 5 min and we averaged the data with a temporal resolution of 2 hours.

The dry aerosol scattering coefficients at 525 nm were measured simultaneously to represent air pollution condition by an integrated

nephelometer (Ecotech Pty Ltd., Aurora 3000) with flowrate of 3 L/min and temporal resolution of 1 min. Similar to the measured

BCPMSD, aerosol scattering coefficients were also averaged with a temporal resolution of 2 hours.

For AE51, the influence of loading effect was resolved by using $\sigma_{ab,corrected} = (1 + k \cdot ATN)\sigma_{ab,uncorrected}$. $\sigma_{ab,corrected}$ and

$\sigma_{ab,uncorrected}$ are the corrected and uncorrected $\sigma_{ab}$, respectively. Factor $k$ is set as 0.004 (Zhao et al., 2019b) and $ATN$ is the

measured light attenuation by particles collected on the filter of AE51. A recommended compensation of 2.6 for AE51 is introduced

here to mitigate the multiple scattering problem (Zhang et al., 2018). Zhao et al. (2019b) added AE33 (3 L/min) to measure the bulk

EBC simultaneously while measuring BCPMSD. The bulk EBC from AE33 and the integrated EBC from BCPMSD measured by

AE51 were then compared. Results showed that the variation trends and magnitudes of EBC measured by AE33 and AE51 were in

good consistence. Therefore, the BCPMSD measured by AE51 was regarded as the measurement results of AE33, and the size-

resolved $\sigma_{ab}$ were retrieved by the constant MAC value of 7.77 m$^2$/g used in AE33. The traditional BCPMSD is retrieved from

AE51-measured size-resolved $\sigma_{ab}$ with a constant MAC value of 7.77 m$^2$/g used for AE33. In our new method, the MAC is variable

as a function of BC core size (D$_{BC}$) and particle diameter (D$_{particle}$).

For scattering correction, a scattering correction factor $C_f$ is required to account for the scattering effect of the filter

matrix: $\sigma_{ab,corrected} = \sigma_{ab,uncorrected}/C_f$. $C_f$ is determined by simultaneous measurement of $\sigma_{ab}$ by PASS-3 ($\sigma_{ab,PASS-3}$) and

AE33 ($\sigma_{ab,AE33}$). $\sigma_{ab,PASS-3}$ is considered as $\sigma_{ab,corrected}$ and $\sigma_{ab,AE33}$ is considered as $\sigma_{ab,uncorrected}$. The wavelengths of

PASS-3 and AE33 are not the same. The measured wavelengths of AE33 (370 nm, 470 nm, 520 nm, 590 nm, 660 nm, 880 nm and

950 nm) were interpolated to the measured wavelengths of PASS-3 (405 nm, 532 nm, and 781 nm). Specifically, For AE33, 405 nm,

532 nm and 781 nm are calculated at wavelengths pairs of (370 nm, 470 nm), (520 nm, 590 nm) and (660 nm, 880 nm) through

Ångström relationship:

$$\frac{\sigma_{ab}(\lambda_1)}{\sigma_{ab}(\lambda_2)} = \left(\frac{\lambda_1}{\lambda_2}\right)^{-\alpha_{ab}}.$$

More detailed information can be found in Zhao et al. (2020). Measurement results at Taizhou and Beijing showed that all the ratios

of $\sigma_{ab}$ measured by AE33 with a measurement flowrate of 3 L/min and PASS-3 with flowrate of 1 L/min at the three wavelengths

varied slightly over the East and North China Plain ($\pm$ 0.04), with an average value of 2.9. Therefore, as the measurement results

between AE33 and AE51 were consistent, all the size-resolved $\sigma_{ab}$ from AE51 adopted in this study were corrected with $C_f = 2.9$.

For the SP2 system, the aerosol samples were analyzed in SP2 (0.12 L/min) to identify the BC-containing particles and in CPC

(0.28 L/min) to count the total number of particles. When a BC-containing particle travels through the laser beam (1064 nm) inside

the SP2, it emits incandescent light. The avalanche photodetectors (APDs) around the laser beam can detect the incandescence

signal. Then the BC-containing particle is detected. N$_{BC}$ can be determined as the ratio of the number of BC-containing particle to

that of total aerosol particle. Detailed configuration of the SP2 system has been demonstrated in a previous study (Zhao et al., 2019a).

According to the measurements at Taizhou, only 17% of the ambient particles contained BC averagely for bulk aerosol populations.

All the measurement systems at the three sites are shown in Fig. S1 in the supplement.

**3 Method**

For current filter-based intruments, EBC are generally derived from $\sigma_{ab}$ under assumption of a constant MAC value. However, the
MAC values are enhanced by different degrees when BC particles are mixed with other weakly-absorbing materials, leading to large
uncertainties on EBC retrieval. In order to gain more accurate EBC, it is critical to consider the discrepancies in MAC caused by
variations in the coating process, BC sizes, etc. Among with the core-shell configuration hypothesis, developing the relationship
between MAC, $D_{BC}$, and $T_{shell}$ is a new approach to correlate EBC with $\sigma_{ab}$.

**3.1 Core-shell geometry of aerosol particles**

To evaluate the theoretical discrepancies in MAC values caused by the corresponding impact factors, a proper model is required to
simulate the optical properties of BC-containing particles to a good approximation. Three widely employed mixing states are used
to represent the structure of BC-containing aerosols: internal, external, and core-shell model (Ma et al., 2011;China et al., 2015).
Generally, freshly-emitted BC particles are chain-like aggregates of small spheres. During the coating process, the chain-like BC
aggregates become more compact as they collapse and are coated as a core by organic and inorganic materials (Bond and Bergstrom,
2006). Therefore, core-shell configuration is more plausible (Jacobson, 2000). Ma et al. (2012) also indicated that the core-shell
assumption can provide a better performance in optical closure than the internal or external models. Furthermore, Moffet et al. (2016)
studied particle mixing state and morphology using scanning transmission X-ray microscopy and highlighted that core-shell
structure dominated the mixing state of ambient aerosol particles. As aerosols are assumed to be core-shell mixed, with a spherical
BC core in the center of the coating sphere. Therefore, the Mie model was used to simulate the optical properties of BC particles
with core-shell mixing state. The consistency in observed and theoretical values obtained using Mie and core-shell morphology
support the suitability of this method (Cappa et al., 2012).

**3.2 Simulation of MAC for BC-containing particle using Mie theory**

Many optical simulations for BC particles with concentric sphere geometry have been reported and the corresponding results show
that the absorption of a pure BC particle will be enhanced when a shell composed of non-absorbing material deposits on this pure
BC particle. Since the optical properties were focused on rather than chemical compositions of the mixed aerosols, a simplified
hypothesis of BC/sulfate mixtures, which is common in the atmosphere (Khalizov et al., 2009), was introduced in the algorithm for
calculating EBC.
The reason of AE33 using 880 nm to determine EBC is that aerosol absorption at 880 nm is mainly from BC (Ramachandran and
Rajesh, 2007). At shorter wavelength, absorption of organic carbon is not negligible any more, leading to difficulty of extracting
BC absorption from total aerosol absorption. Therefore, MAC at 880 nm is discussed in this study and the MAC distribution for a
wide range of core and coating sizes at the wavelength of 880 nm are simulated with Mie scattering theory. The refractive index
(RI), dependent on light wavelength, is an important parameter to determine aerosol optical properties. However, due to different
sources of BC, both the real and imaginary part of RI varies over a significantly wide range. Liu et al. (2018) summarized RI values
for specific wavelengths and showed that the real part is generally in the range of 1.5 to 2.0 while the imaginary part usually varies
from 0.5 to 1.1 (Sorensen, 2001;Bond and Bergstrom, 2006). Therefore, the real part and imaginary part of RI were set to change
from 1.5 to 2.0 and from 0.5 to 1.1, respectively, with a step increase of 0.01. Meanwhile, the RI of sulfate was set as $1.55\text{-}10^{-6}i$ and
the density of BC was set as 1.8 g/cm$^3$, similar to Bond et al. (2006). A total of 3111 values were obtained, and the mean values are
illustrated in Fig. 1. The $D_{BC}$ and total aerosol particle diameter ($D_{particle} = D_{BC} + T_{shell}$) ranged from 10 to 700 nm.
Figure 1 presents several features of the variation pattern of MAC at 880 nm. MAC values varied significantly with $D_{BC}$ and the
thickness of non-absorbing coating, which indicated that light absorption of BC-containing particles was sensitive to the BC core
and the coating. When the $D_{BC}$ was less than 100 nm, the thickness of the coating dominated the variation of MAC values, and MAC
values increased with increasing $T_{shell}$. As the $T_{shell}$ increased, the lensing effect became more significant, the light absorption
consequently also increased with increasing $T_{shell}$. MAC value can increase from 4 m$^2$/g to about 17 m$^2$/g when the total aerosol size
reached up to 700 nm, which indicated that light absorption can be enhanced significantly by the coating. When the $D_{BC}$ was larger
than about 100 nm, both $T_{shell}$ and $D_{BC}$ determined MAC values and $D_{BC}$ played a more important role considering that the majority
of the contour lines tilted to the axis of particle diameter. MAC increased with increasing $T_{shell}$ and decreased with increasing $D_{BC}$.
At this range ($D_{BC} > 100$ nm), the coating still enhanced absorption. For pure BC particle, MAC decreased with increasing $D_{BC}$
when $D_{BC} > \sim 220$ nm, which indicated that the absorption of large BC particles was less than that of small BC particles per unit
mass. If the $D_{particle}$ or the coating ($T_{shell}$) was fixed, larger $D_{BC}$ generally corresponded to a smaller MAC. Not only did the MAC of
coated BC-containing particle vary significantly, but the variation of MAC of pure BC particle was also nonnegligible. For smaller
pure BC particle, the MAC increased slightly with BC size until $D_{BC}$ reached 220 nm. Then, MAC decreased with increasing $D_{BC}$.
Therefore, light absorption can be significantly influenced by coating state, and a constant MAC value of 7.77 m$^2$/g used in AE33
is only appropriate for a very limited condition.
**3.3 New method to retrieve EBC by considering the variation of MAC**
In this subsection, a new method is introduced to determine EBC from the measured $\sigma_{ab}$. At a given $D_{particle}$ ($=D_{BC} + T_{shell}$) selected
by DMA, if $D_{BC}$ is prescribed, the corresponding $T_{shell}$ is determined. Combining the simultaneously measured particle number
concentration ($N(D_{particle})$) by CPC downstream the DMA and the prescribed $N_{BC}$, the number of BC-containing particles
($N_{BC}(D_{particle})$) is then determined. $\sigma_{ab}$ can then be calculated by Mie model with $D_{particle}$, $D_{BC}$ and $N_{BC}(D_{particle})$. If the calculated
$\sigma_{ab}$ matches measured $\sigma_{ab}$ by AE51, then the prescribed $D_{BC}$ is considered as diameter of BC core at $D_{particle}$. Else, $D_{BC}$ is changed
until calculated $\sigma_{ab}$ equals measured $\sigma_{ab}$. MAC can be calculated by Mie model with $D_{particle}$, $D_{BC}$ and a presumed BC density.
EBC at $D_{particle}$ is then derived by dividing measured $\sigma_{ab}$ by MAC. BCPMSD can then be derived through changing $D_{particle}$ selected
by DMA.
As the absorption properties of BC particles in different coating states have been evaluated with the Mie model, as shown in Fig. 1,
a simplified algorithm was proposed for deriving BCPMSD through a pre-calculated look-up table. For each $D_{particle}$ selected by
DMA, if a $D_{BC}$ is assumed, the corresponding MAC of the particle can be derived from the look-up table. Then, the $\sigma_{ab}$ can be
derived from the MAC, the assumed BC density (1.8 g/cm$^3$ in this study), and $N_{BC}$ (17% for each $D_{particle}$). We adjusted the guessed
$D_{BC}$ until the difference between calculated and measured $\sigma_{ab}$ was within an acceptable range (0.1%). Consequently, the $D_{BC}$ and
thus the EBC at $D_{particle}$ was determined. The EBC at different aerosol sizes were derived separately. Finally, the size-resolved EBC
and the bulk EBC were obtained. The iterative procedure is illustrated in Fig. 2.
It should be pointed out that the retrieval algorithm of BCPMSD is based on the assumption that BC-containing particles of a fixed
diameter are all core-shell mixed and the corresponding $D_{BC}$ for a specific $D_{particle}$ is same. The uncertainties caused by using
idealized core-shell model was discussed in section 5.1. A constant number percentage (17%) of BC-containing particles was
adopted in this study. However, the BC-containing particle fraction varied with the primary source, time, coagulation, and extent of
atmospheric process. The influence attributed to the constant fraction of BC-containing particles was discussed in section 5.2.
Additionally, Bond et al. (2013) summarized the density for different graphitic materials. The density values are $1.8 – 2.1$ g/cm$^3$ for
pure graphite, $1.8 – 1.9$ g/cm$^3$ for pressed pellets of BC, and $1.718$ g/cm$^3$ for fullerene soot. A constant density ($1.8$ g/cm$^3$) for BC
was briefly used to calculate MAC and BC mass from the volume of particles with a diameter of $D_{BC}$. Therefore, the uncertainty of
derived EBC in this study simply depends on the ratio of $1.8$ g/cm$^3$ and the real density. Finally, the MAC values in the look-up
table were the mean values for different RI and the corresponding effects were discussed in section 5.3.
**4 Results and discussion**
Figure 3 provides a comprehensive overview of the variations in measured and retrieved size-resolved parameters during the
campaign. As evident from Fig. 3(a), for the BCPMSD derived by the new method, two modes were found. Figure 4(a) shows the
averaged BCPMSD derived from the new method and AE33 during the campaign. The fine mode was located between 97 – 240
nm while the coarse mode was located between 240 – 602 nm. Figure 3(b) represents the relative deviations between the BCPMSD
derived from the newly proposed method and those derived from a constant MAC value of 7.77 m$^2$/g at 880 nm. The results show
that there exist two opposite deviation trends before and after the turning point around 280nm. For aerosol particles larger than 280
nm, the EBC derived by the new method were mostly lower than those derived with the constant MAC value of 7.77 m$^2$/g at 880
nm. In contrast, when aerosol particles were smaller than 280 nm, the EBC from the new method were significantly higher than
those calculated by the constant MAC, as shown in Fig. 3(c). Figure 3(c) shows the time series of size-resolved MAC during the
derivation process of BCPMSD. According to Fig. 3(c), for aerosol particles smaller than 280 nm, the corresponding MAC was
almost lower than 7.77 m$^2$/g at 880 nm. This is because the MAC values of particles smaller than 280 nm are mostly lower than
7.77 m$^2$/g, as represented in Fig. 1. A smaller MAC implies a weaker absorption ability, which means that the same measured $\sigma_{ab}$
will correspond to an increased EBC. Therefore, more EBC were derived from the new method. For aerosol particles larger than
280 nm, in order to match the measured $\sigma_{ab}$, the corresponding $D_{BC}$ were generally found to be in those regions of look-up table
where the MAC values were larger than 7.77 m$^2$/g at 880 nm (Fig. 3(c)). Thus, the BC mass loadings for particles larger than 280
nm were found to be less than those calculated with the constant MAC value of 7.77 m$^2$/g at 880 nm. From Fig. 3(c), it can be seen
that MAC varied from less than 4 m$^2$/g to larger than 10 m$^2$/g at 880 nm, which implies a large variability of the absorption ability
of BC-containing particle. Therefore, if the conversion between EBC and $\sigma_{ab}$ is required, the consideration of variation in mixing
state is highly recommended. The simultaneously measured scattering coefficients at 525 nm were introduced here to represent air
pollution. As shown in Fig. 3(d), the observation station experienced different levels of pollution. Deviations of EBC derived from
the newly proposed method and the constant MAC at different aerosol sizes did not show dependencies on pollution conditions.
Figure 3(e) shows the time series of EBC at fine and coarse modes. The EBC were more concentrated in the fine mode than in the
coarse mode. The EBC at fine mode were found to be higher than those at the coarse mode for 73% of the campaign duration. The
variation trends of bulk EBC calculated by considering the variations of MAC and a constant MAC were similar (Fig. 3(f)). The
bulk EBC calculated by the new method were higher than those derived by the constant MAC in 83% of the campaign duration.
The EBC calculated from the new method and AE33 for different aerosol size ranges were statistically analyzed. As shown in Fig.
4, for all EBC of aerosols ranging between 97 – 602 nm and 97 – 280 nm derived from new method and AE33, strong linear
relationships were observed with correlation coefficients of 0.99 and 1.00, respectively. The ratios between the EBC derived from
AE33 and the new method for aerosol diameter ranges of 97 – 602 nm and 97 – 280 nm were 0.84 and 0.69, respectively, indicating
that the EBC obtained from AE33 was 16% lower for bulk aerosol particles and 31% lower for aerosols smaller than 280 nm. For
the diameter range of 280 – 602 nm, MAC values varied significantly and the deviations in EBC derived from the new method and
AE33 were divided into two types with a boundary of 0.7 $\mu g/m^3$. If the EBC derived from AE33 was lower than 0.7 $\mu g/m^3$, there
was a relatively consistent ratio of 1.13 between the EBC from the new method and AE33, with a correlation coefficient of 0.95.
Therefore, BC mass loading from the AE33 algorithm was 13% higher for aerosol particles larger than 280 nm and EBC lower than
0.7 $\mu g/m^3$. However, when the EBC derived from AE33 was larger than 0.7 $\mu g/m^3$, data points become discrete, and the relationship
between the EBC derived from AE33 and the new method could be expressed through an equation (y = 0.29 + 0.48x). However,
these comparisons for aerosols at different size ranges were obtained based on the measurements in the NCP. Additionally, the
number of samples where EBC of 280 – 602 nm were larger than 0.7 $\mu g/m^3$ was too small. Further studies on BCPMSD in
conjunction with the PNSD measurements at different sites need to be carried out.
**5 Uncertainty analysis**
**5.1 The uncertainties of MAC caused by using idealized core-shell model**
An idealized concentric core-shell model with a spherical BC core fully coated by sulfate was configured to study the MAC of BC
aerosols and derive the EBC in this study. However, freshly emitted BC particles were found to normally exist in the form of loose
cluster-like aggregates with numerous spherical primary monomers (Liu et al., 2015). Soon after, these aggregates become coated
with other components and collapsed to a more compact form during the coating process (Zhang et al., 2008;Peng et al., 2016).
Therefore, the uncertainty in the idealized core-shell configuration is discussed in this subsection.
**5.1.1 The formation of BC aggregates with a determined morphology**
The fractal aggregates of BC have been well described by fractal geometries through the well-known statistical scaling law
(Sorensen, 2001):

$$N = k_f \left(\frac{R_g}{a}\right)^{D_f},$$

where $N$ is the number of "same-sized" monomers in the cluster, $a$ is the monomer radius, $D_f$ and $k_f$ are known as the fractal
dimension and fractal prefactor respectively, determining the morphology of BC cluster. The compactness of a fractal aggregate
increses with increasing $D_f$ or $k_f$. $R_g$ is the gyration radius, infering the overall aggregate radius, determined by
$$R_g = \sqrt{\frac{1}{N}\sum_{i=1}^{N} r_i^2},$$
where $r_i$ represents the distance of the $i$-th monomer from the center of mass of BC cluster.
In order to generate fractal-like aggregates with given $N$, $R_g$, $a$, $D_f$ and $k_f$, the sequential algorithm proposed in Filippov et al.
(2000) is introduced in this study to add the primary monomers one by one. On condition that there is an aggregate including $N-1$
monomers, the $N$-th monomer is constantly placed randomly until it has at least one contact point with the previously attached
$N-1$ monomers with no overlapping. Besides, the mass center of the next $N$-th monomer must obey the rule as follows:
$$(r_N - r_{N-1})^2 = \frac{N^2 a^2}{N-1}\left(\frac{N}{k_f}\right)^{2/D_f} - \frac{Na^2}{N-1} - Na^2\left(\frac{N-1}{k_f}\right)^{2/D_f},$$
where $r_{N-1}$ and $r_N$ are the mass center of the first $N-1$ monomers and the $N$-th monomer, respectively. After the fractual
configuration of BC aggregates, the absorption properties of BC containing particles need to be evaluate.
The fractal dimensions for aged BC aggregates are generally close to 3 (Kahnert et al., 2012). The aim of this study is to evaluate
the effects of aerosol microphysics on the absorption enhancement of fully coated BC particles, which can be regarded as the aged
BC aerosols. Therefore, the fractal dimension $D_f$ is set to be 2.8 and $k_f$ is generally set to be 1.2. The diameter of the primary
monomers is usually between 20-50 nm and the number of the primary monomers for an aggregates is between 50-300. The size of
BC core calcuated by the new method is smaller than 300 nm most of the time during Zhangqiu campaign. The diameter of primary
monomers is set to be 50 nm and the number of the primary monomers for an aggregates ranges from 2 to 200, leading to the largest
size of volume equivalent BC core close to 300 nm. The real part of BC is generally in the range of 1.5 to 2.0 while the imaginary
part usually varies from 0.5 to 1.1 (Liu et al., 2018). Therefore, the mean value 1.75 for BC real part and 0.8 for BC imaginary part
are adopted here to calculate MAC values for BC/sulfate mixtures at the wavelength of 880 nm.
**5.1.2 Multiple Sphere T-matrix (MSTM) method**
As the traditional Mie model is not available for the fractal aggregates, the widely used MSTM method is employed here to quantify
the absorption properties of BC clusters (Mackowski and Mishchenko, 1996;Mackowski, 2014). The addition theorem of vector
spherical wave functions is used in MSTM method to describe the mutual interactions among the system. The T-matrix of aggregates
used to derive particle optical properties can be obtained from these individual monomers. MSTM method can calculate light
scattering and absorption properties of the randomly oriented aggregates without numerical averaging over particle orientations if
the position, size and refractive index of every spherical element are given. However, the MSTM method is only applicable to
evaluate the aggregates of spheres without overlapping and it is carried out with high computational demand.
The deviations showed in Fig. 5 are derived by subtracting MAC values calculated by MSTM model by those calculated by Mie
model. The results show that most of the MAC values calculated by assuming BC particles in the form of cluster-like aggregates
are smaller when the size of BC core is smaller than 150 nm and the overall deviation is within 4 %, which indicates that Mie theory
is a good approximation to the BC aggregates even when $D_{BC}$ reaches 200 nm. When BC core is larger than 200 nm, the MAC
values calculated by MSTM model increase with the thickness of shell and will be larger than those derived from concentric core-
shell model. The deviations between MAC calculated by the idealized concentric core-shell model and letting BC particles be in the
form of cluster-like aggregates are overall within 15%.
**5.2 The uncertainties of derived EBC caused by using a constant BC-containing particle fraction**
Figure 6 shows the deviation of BCPMSD calculated from different $N_{BC}$ (8.5%, 17%, 34%). For the newly proposed method, using
a constant $N_{BC}$ does not change the size-resolved distribution mode. There is still a fine mode and coarse mode with a boundary of
240 nm. Besides, the influence of using different $N_{BC}$ to derived EBC is very limited when particles are larger than 200 nm. However,
the deviations between the EBC derived from different $N_{BC}$ are large when particles diameters are smaller than 200 nm. At this
range, if $N_{BC}$ is underestimated, the EBC will be underestimated. On the contrary, the EBC is overeatimated if $N_{BC}$ is overestimated.
**5.3 The uncertainties of MAC caused by variation of RI**
As the RI of BC is still reported to vary over a wide range and the MAC used in this study was a mean value, it is critical to assess
the impact caused by variation in the real and imaginary parts of RI on the calculated MAC and the derived EBC. For aerosol
particles with given $D_{BC}$ and $T_{shell}$, we calculated the MAC of BC with the real part of RI ranging from 1.5 to 2.0 and imaginary part
ranging from 0.5 to 1.1. The step increase of both real and imaginary parts was 0.01 and there were 3111 MAC values for every
aerosol particle with given $D_{BC}$ and $T_{shell}$. The ratio of the standard deviation to the mean value for these 3111 MAC values have
been presented to demonstrate the uncertainty in MAC due to the uncertainty of BC RI.
Figure 7(a) shows the uncertainties in MAC along different values of $D_{BC}$ and $T_{shell}$. It shows that aerosol particles with small BC
core have larger uncertainties and all the uncertainties were below 24%, implying a large variation in absorption for BC-containing
particles with small BC core. When $D_{particle}$ was fixed, the uncertainties decreased with increasing $D_{BC}$. When $D_{BC}$ was determined,
the uncertainties did not change much with $T_{shell}$, indicating the importance to quantify $D_{BC}$ for BC-containing particles in order to
reduce RI-related uncertainty in absorption. For pure BC particles, the uncertainties also decreased with increasing BC particle size
significantly from over 22% at 100 nm to less than 2% at 600 nm. Figure 7(b) shows the uncertainties when the imaginary part was
fixed at 0.8 and the real part ranged from 1.5 to 2.0 with an interval of 0.01. It can be seen that when the imaginary part of RI was
fixed, variations in the real part led to slight uncertainties. All the uncertainties were found to be below 14%. Figure 7(c)
demonstrates the uncertainties when the real part was fixed at 1.75 and the imaginary part ranged from 0.5 to 1.1 with an interval
of 0.01. Comparing Fig. 7(a) and 7(c), we can see that the patterns of MAC uncertainties were similar. Overall, the uncertainties
were dominated by the variations of the imaginary part and only slightly affected by variations in the real part. Therefore, it is highly
recommended to reduce the uncertainties in the imaginary part for a more precise absorption measurement.
The variation of EBC caused by the uncertainties in RI were further evaluated. As stated in section 3.2, for a MAC (880 nm) point
at ($D_{particle}$, $D_{BC}$) of Fig. 1, it is a mean value averaged with respect to both real part of RI varied from 1.5 to 2.0 and imaginary part
of RI varied from 0.5 to 1.1. The mean MAC (880 nm) plus corresponding standard deviation (MAC + Std) and minus corresponding
standard deviation (MAC – Std) were used to show the uncertainties in EBC caused by variation of BC RI. As we can see from Fig.
8(a), irrespective of the MAC was MAC + Std or MAC – Std, there was no change in the mode of BCPMSD. The derived EBC of
all particles ranging from 97 – 602 nm increased when MAC – Std was used and decreased when MAC + Std was used. Compared
to the bulk EBC derived by mean MAC, those derived by MAC – Std were higher within 35% (Fig. 8(b)). The decrease in the
derived EBC caused by MAC + Std was less than the increase in the derived EBC caused by the MAC – Std. For both fine and
coarse mode particles, the deviations in EBC caused by MAC + Std or MAC – Std were also within 35% (Fig. 8(c) and Fig. 8(d)).
This sensitivity study indicates that the accuracy of the derived BCPMSD is sensitive to the accuracy of MAC, especially when the
actual MAC is less than the mean MAC.
**6 Conclusions**
There was a significant variability in the MAC values of BC with the size of BC core and the thickness of coating, which exerted a
significant influence on the optical method for deriving EBC. In this study, a new method was proposed to derive EBC considering
the lensing effect of core-shell structure and the consequent MAC variations in MAC.
A look-up table describing the variations of MAC at 880 nm attributed to the coating state and size of BC core was established
theoretically using Mie simulation and assuming a core-shell configuration for BC-containing aerosols. The MAC at 880 nm varied
significantly with different sizes of core and shell from less than 2 $m^2/g$ to over 16 $m^2/g$, indicating a large variation in absorption
ability for BC-containing particle. Then, the EBC at different aerosol sizes were derived by finding an appropriate BC core
configured with a MAC value from the look-up table to close the calculated and measured $\sigma_{ab}$.
This newly proposed method was applied to a campaign measurement in the NCP. There were two modes for BCPMSD at the
accumulation mode separated by 240 nm. For 73% of the cases, the EBC of the fine mode were larger than those of the coarse mode
during the measurement. The EBC derived by the new method were mostly lower than those derived by a constant MAC of 7.77
$m^2/g$ for particles larger than 280 nm, and higher for particles smaller than 280 nm. Similarly, the bulk EBC accumulated from
BCPMSD derived from the new method were mostly higher than those derived from a constant MAC of 7.77 $m^2/g$.
Uncertainty analysis was carried out with respect to assumptions used in this study. The uncertainty caused by idealized core-shell
model was analyzed by substituting the core with cluster-like aggregates using MSTM method, and the resulting relative
uncertainties were within 15%. The uncertainties caused by using a constant number fraction of BC-containing particle was analyzed
by halving and doubling its value, and the results showed that particle larger than 200 nm was insensitive to the number fraction of
BC-containing particle, whereas, for particle smaller than 200 nm, the EBC would be underestimated if the BC-containing particle
fraction was underestimated. The uncertainty in derived EBC that was caused due to the wide range of RI of the BC core was also
studied. The results indicated that the uncertainty of the imaginary part results in larger uncertainties to the MAC compared with
the real part. The relative uncertainty of the derived EBC was within 35%.
This study provides a new way to derive EBC from $\sigma_{ab}$ for the widely-used filter-based measurements. This research deepens our
understanding of the uncertainty in measured EBC caused by the utilization of a constant MAC and illustrates the great necessity to
take the variation of MAC into account. The new method improves the measurements of BCPMSD and deepens the understanding
about the significant influence of mixing state on the absorption of BC.
**Data availability**
The measurement data involved in this study are available upon request to the authors.

## Author contributions

CZ determined the main goal of this study. WZ and WT designed the methods. WZ carried them out and prepared the paper with contributions from all co-authors.

## Competing interests

The authors declare that they have no conflict of interest.

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

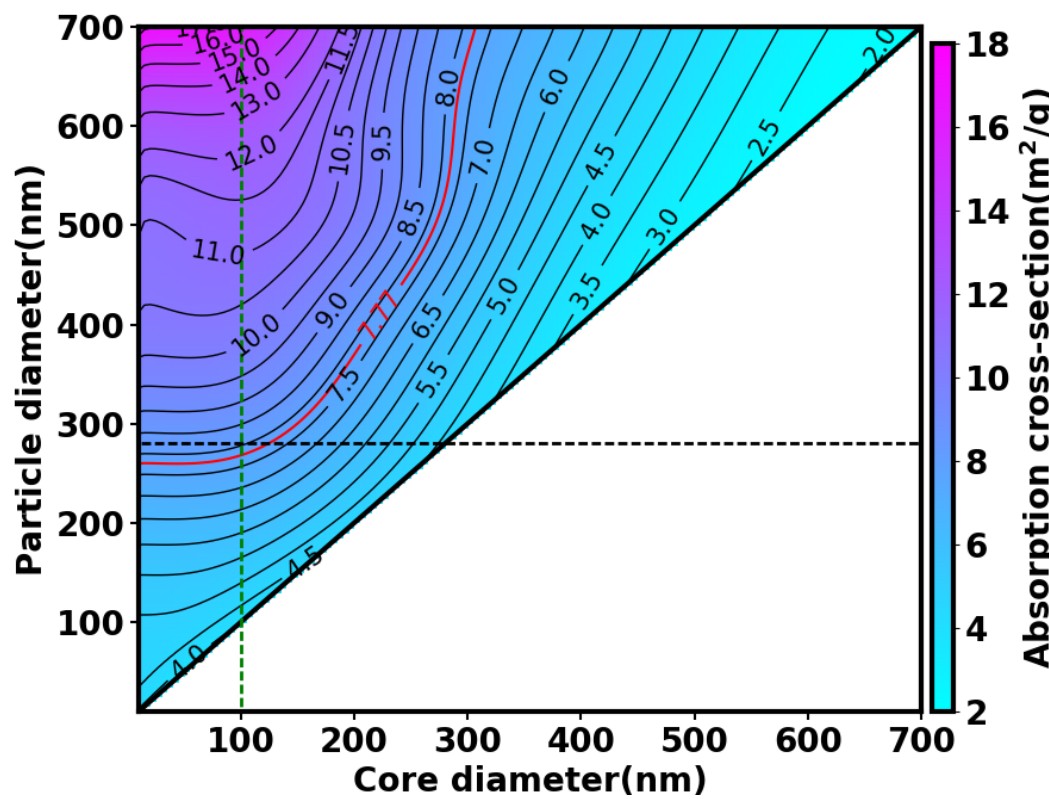

**Figure 1. Variations in MAC as a function of $D_{BC}$ and $D_{particle}$, calculated by the concentric core-shell Mie model at the**
**wavelength of 880 nm. The red solid line is the constant MAC value used in AE33. The bold black solid line is the 1:1 line**
**that presents the variations in MAC for pure BC particles with different $D_{BC}$. The horizontal black dashed line distinguishes**
**particles with a diameter of 280 nm while the vertical green dashed line indicates a $D_{BC}$ of 100 nm.**

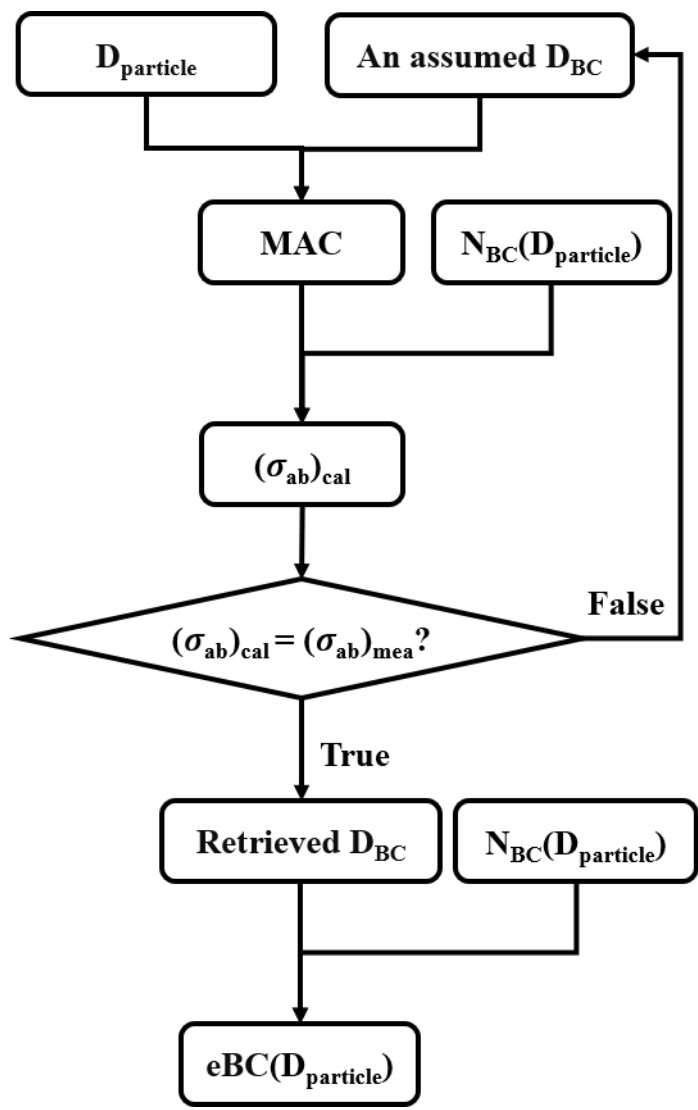


**Figure 2. Schematic diagram of the iterative algorithm for retrieving the EBC at a fixed particle diameter based on the look-up table of MAC, particle size and core size. $(\sigma_{ab})_{cal}$ and $(\sigma_{ab})_{mea}$ represent calculated and measured absorption coefficients, respectively. $N_{BC}(D_{particle})$ indicates the number concentration of particle containing BC at the given $D_{particle}$.**


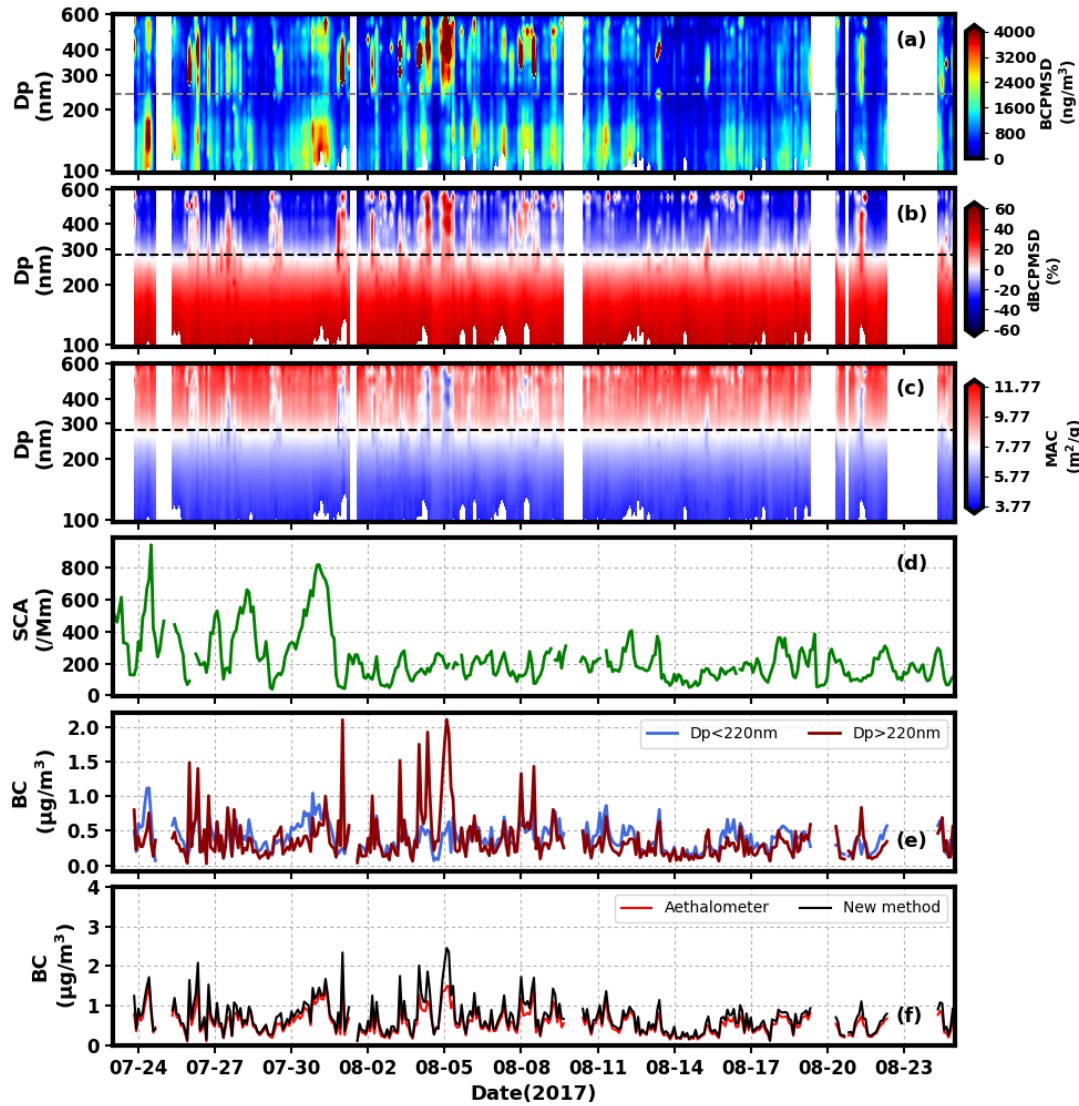


**Figure 3. Time series of (a) the BCPMSD derived from the new method proposed in this work. The dashed line indicates the**
**particle size of 240 nm; (b) relative deviations between BCPMSD derived from the new method (varied MAC) and constant**
**MAC at 7.77 m²/g. The dashed line indicates the particle size of 280 nm; (c) the size-resolved MAC. The dashed line indicates**
**the particle size of 280 nm; (d) the scattering coefficients at 525 nm; (e) the EBC integrated for particles smaller than 220**
**nm (blue) and larger than 220 nm (red); and (f) the EBC determined by the new method (black) and constant MAC of 7.77**
**m²/g (red).**

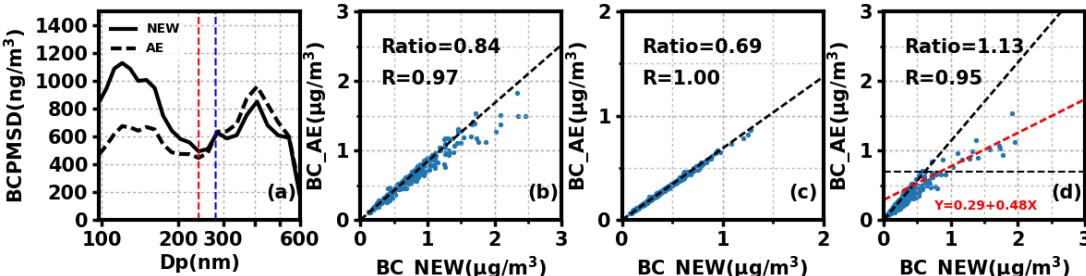


**Figure 4. Comparison between the EBC derived from the new method and from constant MAC of 7.77 m²/g used by AE33.**
**(a) shows the results of the BCPMSD, the dashed black line is the result of constant MAC of 7.77 m²/g used by AE33, and**
**the solid black line represents the results from the new method; the dashed red line is the split line of 240 nm between fine**

mode and coarse mode, and the dashed blue line is the split line of 280 nm between the opposite tendencies of deviations in

BCPMSD; (b) the bulk EBC integrated from 97 nm to 602 nm; (c) the fine mode EBC integrated from 97 nm to 280 nm; (d)

the coarse mode EBC integrated from 280 to 606 nm; the dashed black line represents boundary of 0.7 μg/m³ and the red

dashed line is the regression line for the EBC larger than 0.7 μg/m³.

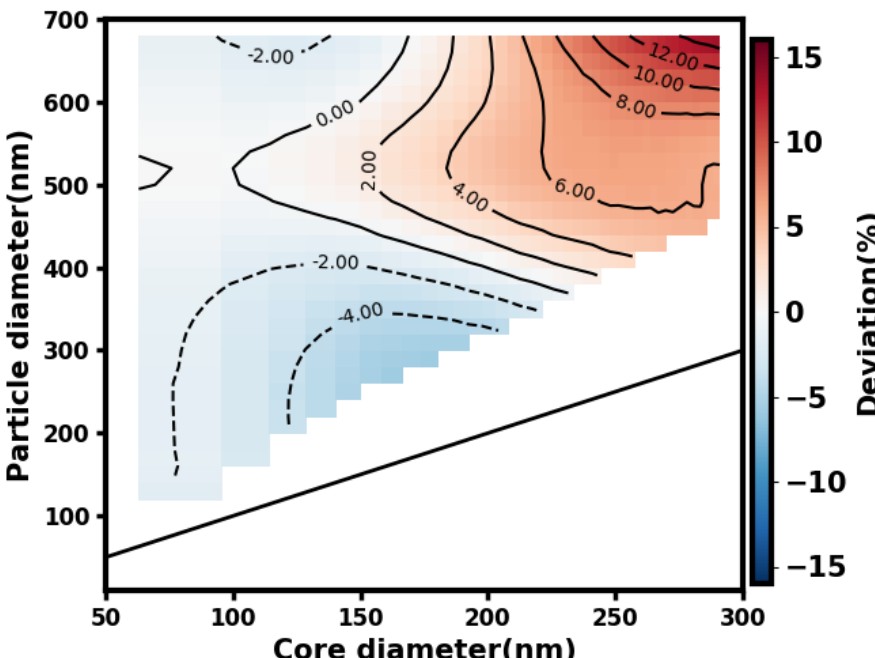

Figure 5. Relative deviations of MAC values calculated by idealized concentric core-shell model and letting BC particles be

in the form of cluster-like aggregates. The solid line is 1:1 line.

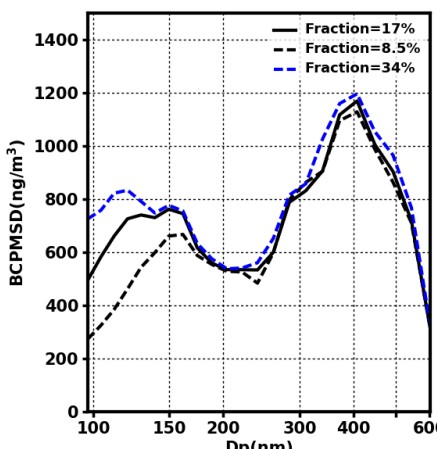

Figure 6. The derived BCPMSD by using different constant BC-containing particle fraction. The solid black line represents

the result derived from a fraction of 17%. The dashed black line and blue line show the results derived from a fraction of

half of 8.5% and double of 34%.

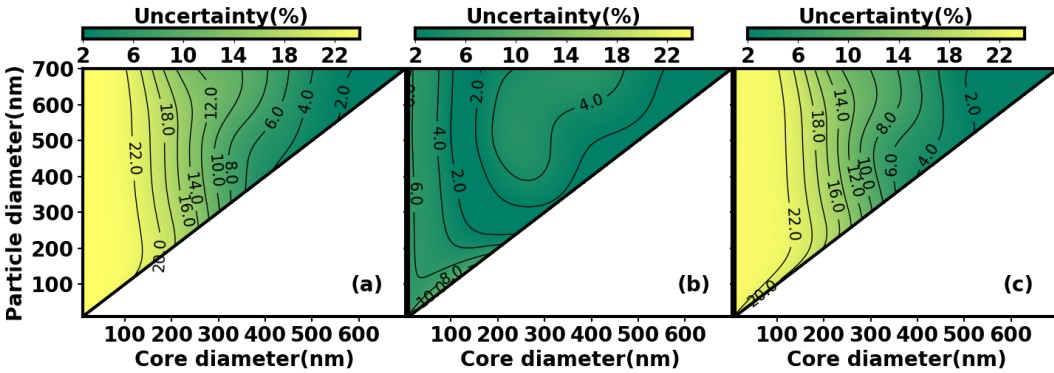

Figure 7. Uncertainty in MAC of BC when (a) real part of RI ranges from 1.5 to 2.0 and imaginary part ranges from 0.5 to

1.1; (b) real part of RI ranges from 1.5 to 2.0 and imaginary part is fixed at 0.8 and (c) real part of RI is fixed at 1.75 and

imaginary part ranges from 0.5 to 1.1. The bold black solid line is the 1:1 line and presents the uncertainty of MAC for pure

BC particles with different RI.

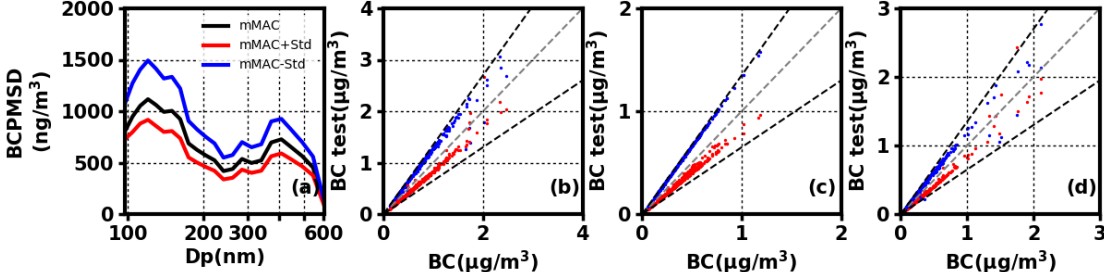

Figure 8. (a) The BCPMSD calculated by using the look up table with mean MAC (black line), mean MAC plus the

corresponding standard deviation (red line) and mean MAC minus the corresponding standard deviation (blue line); the

EBC derived by the look up table with mean MAC versus those derived by the look up table with mean MAC plus standard

deviation (red dots) or mean MAC minus standard deviation (blue dots) for (b) aerosol particles ranging from 97–602 nm;

(c) aerosol particles ranging from 97–240 nm (fine mode); and (d) aerosol particles ranging from 240–602 nm (coarse mode).

The dashed black line represents the 35% deviation from the 1:1 line (dashed grey lines).