# Peer review of "Determination of equivalent black carbon mass concentration from"

_Atmospheric Measurement Techniques, 2020_

## Referee Comment (RC1) · Anonymous Referee #1 · 12 Oct 2020

A new method by considering the variation in MAC is developed to obtain BC mass size distribution and then bulk BC mass concentration from size-resolved light absorption measurements. Size-resolved MAC calculated on the basis of core-shell Mie model is mainly discussed, which is determined by Dp-dependent DBC and coating thickness. However, there are many assumptions in calculation processes, e.g., same DBC and coating thickness at each selected mobility size, a constant number fraction of BC-containing particles, etc. Meanwhile, measurements were not described clearly.

The significance of this study should be also strengthened. In my point of view, compared to BC mass loading, the light absorption measurements are more required to

evaluate the influences of BC particles on solar radiation. Thus, MAC is likely to be more important for converting bulk BC mass loading, which can be directly measured by using chemical method (e.g., Thermo Optical Reflection-EC) or laser-induced incandescence techniques (e.g., SP2-rBC), to light absorption in climate research. The current study is more important for obtaining BC mass size distribution from size-resolved absorption measurement. BC mass size distribution obtained from the DMA-AE51 measurement based on the new method is also suggested to compare with that obtained from the direct measurement from DMA-SP2 system, which has used in the field campaign.

Furthermore, the Mie model is likely to not suitable for the calculation of BC aggregates with large sizes. For a small BC particle (core), the mass equivalent diameter of the assumed BC sphere is much smaller than the wavelength (880 nm) resulting in a less effect of morphology to absorption. In this case, the Mie model is somewhat feasible for absorption estimation. However, for a large BC particle (core), its mass equivalent diameter is close to the wavelength (i.e., large size parameter); thus, the absorption is largely influenced by the morphology. Moreover, large BC particles are more likely to exhibit loose fractal aggregates with thin coating, thus, is likely much different from core-shell structure. MAC in this case cannot be well depicted by using Mie model.

In general, some improvements are necessary before the manuscript can be accepted for publication.

Specific comments: 1. Wavelength should be addressed when the absolute value of MAC is mentioned. 2. Line 13, what do the 'different core-sell structures' mean? Different core size and shell thickness? 3. Line 57–58, Bond and Bergstrom (2006) just suggested a consistent MAC for fresh (uncoated) BC particles. 4. Line 73, a more detailed but clear description of BCPMSD measurement should be addressed. From my understanding, major results and discussion presented in this study are based on the BCPMSD measurements (using DMA-AE51?) at Zhangqiu site. DMA-SP2 measurements at Taizhou, and comparisons of AE33 with PASS-3 at Taizhou and Beijing are
mostly used to provide essential parameters (e.g., number fraction of BC-containing particles, multi-scattering correction factor for AE33, etc.) for the BCPMSD retrieval. 5. Line 112–115, the method to determine the size-resolved number fraction of BC-containing particles should be introduced briefly. How to deal with the effect of multi-charged particles in the DMA-SP2 system. Why the number fraction of BC-containing particles at Taizhou can be used to represent that at Zhangqiu? 6. Line 120, why absorption coefficients measured by AE33 are 2.9 times those measured by PASS-3? Does this ratio mean the multi-scattering effect of the filter loading method? However, as mention in line 106, a compensation factor of 2.6 has been introduced to mitigate multiple scattering effect. Was the PASS-3 well calibrated before the measurement? 7. Line 147, although the mantle chemical species would not influence largely the results presented in this study, BC/OM mixtures are more likely existed in the atmosphere of studied regions.

---

## Referee Comment (RC2) · Anonymous Referee #2 · 13 Oct 2020

The study that is discussed in this manuscript is focused on developing a more robust methodology for the derivation of the mass absorption cross section (MAC) that relates the mass of black carbon to its light absorbing properties. As correctly explained by the authors, the MAC can't be a constant such as typically applied by the majority of those users who deploy instruments that measure the light absorption of atmospheric particles. The primary objective of the current study is to derive a MAC that depends on the size of the BC core and thickness of the shell, with some assumptions about the real and imaginary refractive indices of the mixed phase particles. There is nothing fundamentally incorrect about the approach that they take but there are so many problems with the manuscript itself that I am unable to accept this paper without some

major changes as I list below.

1) The authors have ignored the recommendations as proposed by Petzold et al. (2013), recommendations that are generally accepted by the scientific community, on how black carbon (BC) should be reported when derived from instruments that measure light attenuation, i.e. filter based or photoacoustic sensor. BC derived from these techniques should be reported as equivalent BC, or eBC, If or when this paper is resubmitted, the title should reflect clearly that it is eBC that is being discussed, not BC.

2) A large fraction of the introduction is devoted to the importance of BC for climate change due to radiative forcing. What the authors fail to understand is that in the context of their study, the corrections to the MAC that they are proposing is completely irrelevant. Sensors that measure light absorption like the Aethalometer, are already providing the necessary information that is relevant to climate change, i.e. it is not the mass concentration that is important it is the optical cross section. I will address this further below with respect to the mixing state of BC, but the primary point is that the mass concentration of BC is not important when doing radiative transfer calculations if you already have the primary measurements of the coefficients of scattering and absorption. The authors also mention that BC might be efficient CCN or IN, both true statements but again irrelevant with respect to their study. Hence, the introduction needs to be completely rewritten to explain the real relevance of the current study, and that is to set some error bounds on eBC derived from Aethalometer measurements and NOT a cutting edge, new methodology that will in any way improve the accuracy of such measurements.

3) This study should be written up as a detailed analysis of the uncertainties in the MAC related to the mixing state of BC, i.e. the refractive indices, real and imaginary, the wavelength of incident light, and the relative sizes of the core and shell. Secondly, in the introduction, it should be made quite clear how this analysis differs from the many others that have already been published.

4) The methodology that is discussed in this paper is being promoted as a way to derive a more accurate eBC but this is misleading because in order to apply this you need a lot of additional complementary information about the size distribution of the BC, the fraction of particles that are mixed with BC, etc. If you had all the necessary information to begin with, then you wouldn't even need to try and derive eBC using a variable MAC because you would already have enough information to estimate BC without the light absorption instrument. This should be made quite clear in a resubmission of this paper.

5) It is my opinion that the modeling that is being discussed with this study has as much importance for setting the error bars on light absorption derived from the filter based measurements as for setting error bars for deriving eBC. There are many corrections that have been proposed to adjust the light absorption measurements for the impact of overloading, filter matrix effects, etc., but perhaps the results from the current study could also be used to establish how mixed state BC leads to under/over estimates of the absorption coefficient. The authors should give this serious consideration if they want their study to have more relevancy than it does in its current state.

6) The sensitivity studies of how the assumptions impact the results have to be in the main text, not in the supplemental material because these sensitivity studies are critical to the conclusions that are drawn. The current sensitivity studies are too limited and do not take into account all of the assumptions that go into the model. The validity of every assumption must be defended and used to set the limits of the errors eventually reported.

7) I am providing an annotated copy of the PDF with additional comments, questions and suggestions to be addressed by the authors.

Reference Petzold, A., Ogren, J. A., Fiebig, M., Laj, P., Li, S.-M., Baltensperger, U., Holzer-Popp, T., Kinne, S., Pappalardo, G., Sugimoto, N., Wehrli, C., Wiedensohler, A., and Zhang, X.-Y.: Recommendations for reporting "black carbon" measurements, Atmos. Chem. Phys., 13, 8365–8379, https://doi.org/10.5194/acp-13-8365-2013,

2013.

Please also note the supplement to this comment:
https://amt.copernicus.org/preprints/amt-2020-337/amt-2020-337-RC2-
supplement.pdf

―――――――――――――――――

---

## Referee Comment (RC3) · Anonymous Referee #3 · 25 Oct 2020

Black carbon is a very important component in estimation of aerosol's optical properties, which lead to much bias in estimation of its radiation forcing in climate modelling. Because BC is of strong absorption in the solar lights, the share of BC in aerosol is also a reason of uncertainties of remote sensing from radiance-like observation, such as $CO_2$ from satellite measurement in the shortwave infrared wavelength. BC measured by AE51 or other optical instrument, which is easy in operation and used in common, is given under assumption of a constant coefficient of MAC. To correct the differences induced by this assumption, the authors proposed a new method and variant MAC coefficient related to the particles size is used to derive BC. As far as I know, this work is original and very important to derive an accurate BC, and the results from

field measurements and comparisons from different instruments showed here is reasonable, the method may be a good reference in future measurement of BC in the atmosphere. Therefore, I strongly recommend the publish of this work after revision.

My major concern is: 1. As the authors pointed, for the new BC, its shape is chain-like, not a spherical one, so how do you know this method is applicable for the measurement. How many parts of BC is new generated and how many is old one, is there a guess for that? Do you have some samples measured ASAP and others saved and wait some time to let them to be old one? Other minor concerns are related to language and logistics: 2. Line 16, "with in" should be "within" 3. Line 58, what's mean of "degree of MAC" 4. "…Mie model incorporated with core-shell configuration hypothesis was applied in this study to assess the limitation of the constant " should be simplified as ""…Mie model with assumption of core-shell particles was …….." 5. Line 68, "Based on the detailed…." The word "the" should be deleted. 6. Line 73, "The measured BC particle mass size distribution (BCPMSD) was obtained from the field campaign conducted at the Zhangqiu Meteorology Station (36°42'N, 117°30'E), Shandong Province. This field campaign lasted for about 1 month, from July 23, 2017 to August 24, 2017. The Zhangqiu observation site is located in the North China Plain (NCP) and is surrounded by farmland and residential areas, representing regional background conditions of the NCP." Should be rewrite as " The BC particle mass size distribution (BCPMSD) was measured at Zhangqiu Meteorology Station (36°42'N, 117°30'E), Shandong Province, surrounded by farmland and residential areas and a typical site for regional background conditions of North China Plain (NCP). The field campaign lasted for about 1 month, from July 23, 2017 to August 24, 2017." 7. Line 76, the last word "system" should be deleted 8. Line 77, "measurements to determine …." Should be " is used to determine ….." 9. Line 78," The suburban measurement site" , the word "measurement" should be deleted 10. Line 79, the word "the" before "Jianghuai Plain" should be deleted 11. Line 86 and 87, "All the measurements in the three sites were conducted in containers where ambient temperature was controlled within 24 $\pm$ 2 °C with a particle pre-impactor to remove particles larger than 10 $\mu$m

from the input air stream. " should be rewritten as "All the measurements in the three sites were conducted in temperature(24 $\pm$ 2 °C) controlled containers, and a particle pre-impactor is used to remove particles larger than 10 $\mu$m from the input airflow." 12. Line 92, "developed by (Ning et al., 2013). The instrument setup was further improved by Zhao et al. (2019b)." should be "developed by Ning et al. (2013) and improved by Zhao et al. (2019b)" 13. Line 101, "that were used to represent air pollution conditions" should be deleted 14. Line 105, the variables of k and ATN should be italic 15. Line 108, "in this study" should be deleted 16. Beginning of line 115, word "from" should be "at" and the same for line 117 17. Line 117 and 118, " with a measurement flowrate of " should be "with flowrate of" 18. Line 123, "…. through a constant MAC value" should be "under assumption of a constant MAC" 19. Line 130, "an appropriate model simulation is needed for representing a single BC particle's optical properties." What's meaning of this sentence. 20. Line 131, " There are three widely employed mixing states that are used to represent the structure of BC-containing aerosols" should be "Three widely employed mixing states are used to represent the structure of BC-carried aerosols" 21. Line 133, "….chain-like aggregates composed of small spheres" should be "chain-like aggregates of small spheres" 22. Line 139, "the spherical core and shell favor the Mie model" should be deleted. 23. Line 140, "in this study" should be delted. 24. Line 143, could you use other words for the section title ? 25. Line 147, the word "frequent" should be replace by "common" 26. Line 150, "….. at the wavelength of 880 nm, calculated using the Mie theory, has been presented" should be "….. at wavelength of 880 nm are simulated with Mie scattering method." 27. Line 151, "reported to vary with incident light wavelength" should be "dependent on light wavelength" 28. Line 152~153, " as BC particles can be emitted from different fuels and conditions, RI cannot be observed directly, with both real and imaginary part of RI varying over a significantly wide range" should be "due to different source of BC, both the real and imaginary part of RI varies over a significantly wide range" 29. Line 157, "averaged values are illustrated …." Do you mean "mean values……." 30. Please rewrite paragraph between line 168 and 173 to make it simple and clear. 31.

Line 174, the first sentence "The detailed iterative procedure is illustrated in Fig. 2." Should be reposition to the end of last paragraph, and the word "detailed" should be "deleted" 32. Line 175, "represented" should be replace by "shown" 33. Line 175, "a simplified algorithm for deriving BCPMSD was proposed by considering Fig. 1 as a look-up table." Should be rewritten as "a simplified algorithm was proposed to derive BCPMSD through a pre-calculated a look-up table." 34. Line 195 and 196, words "finer mode" and "coarser mode" should be replace by "fine mode " and "coarse mode", please read through the whole draft to replace other similar words. 35. Line 198, "The results indicate that with the boundary of 280 nm, two opposite deviation tendencies exist. " should be replaced by "the results show that there exist two opposite deviation trend before and after the turning point around 280nm" 36. Line 247, "The variations in on…" should be "The variation of …." 37. Line 247, "all MACs in the look-up table in Fig. 1 are the mean values as the imaginary part and real part of BC RI varied over a wide range." What's the meaning of this sentence mean, please rewrite. 38. Please rewrite the whole paragraph between line 247∼260 to make it clear and simple. 39. Line 454 to line 459, please rewrite caption for Figure 3 and make it easy to read. The same the caption of figure 4

Please also note the supplement to this comment:
https://amt.copernicus.org/preprints/amt-2020-337/amt-2020-337-RC3-supplement.pdf

———————————————————

---

## Author Comment (AC1) · 16 Dec 2020

Response to Anonymous Referee #1

**Major comments:**

*1) A new method by considering the variation in MAC is developed to obtain BC mass size distribution and then bulk BC mass concentration from size-resolved light absorption measurements. Size-resolved MAC calculated on the basis of core-shell Mie model is mainly discussed, which is determined by $D_P$-dependent $D_{BC}$ and coating thickness. However, there are many assumptions in calculation processes, e.g., same DBC and coating thickness at each selected mobility size, a constant number fraction of BC-containing particles, etc. Meanwhile, measurements were not described clearly.*

Response: Thanks for your comments. The size resolved MAC in this study was based on core-shell Mie model. The influence of the BC aggregates on the MAC as well as the relative deviation between the core-shell model and BC aggregates were discussed in section 5.1 of the revised manuscript to evaluate the effects of the morphology on MAC. With respect to the assumptions used in this study, their uncertainties were discussed in section 5, such as the uncertainties caused by using idealized core-shell model (section 5.1), by using a constant BC-containing particle fraction (section 5.2) and by variation of refractive index (section 5.3). With respect to description of measurements, a more detailed description of our measurement was added in section 2.

*2) The significance of this study should be also strengthened. In my point of view, compared to BC mass loading, the light absorption measurements are more required to evaluate the influences of BC particles on solar radiation. Thus, MAC is likely to be more important for converting bulk BC mass loading, which can be directly measured by using chemical method (e.g., Thermo Optical Reflection-EC) or laser-induced incandescence techniques (e.g., SP2-rBC), to light absorption in climate research. The current study is more important for obtaining BC mass size distribution from size-resolved absorption measurement. BC mass size distribution obtained from the DMA-AE51 measurement based on the new method is also suggested to compare*

*with that obtained from the direct measurement from DMA-SP2 system, which has used in the field campaign.*

Response: Thanks and we agree with your comments. More sentences were added in this text to stress the significance. The main goal of this study was to derive equivalent BC mass concentration (EBC, after Petzold et al. (2013)) more precisely and obtain BC particle mass size distribution (BCPMSD) from size-resolved absorption measurement. MAC is an important variable that has to be discussed in the process. Derivation of the EBC and related uncertainties were more discussed to emphasize that our goal was to determine EBC more precisely.

*3) The Mie model is likely to not suitable for the calculation of BC aggregates with large sizes. For a small BC particle (core), the mass equivalent diameter of the assumed BC sphere is much smaller than the wavelength (880 nm) resulting in a less effect of morphology to absorption. In this case, the Mie model is somewhat feasible for absorption estimation. However, for a large BC particle (core), its mass equivalent diameter is close to the wavelength (i.e., large size parameter); thus, the absorption is largely influenced by the morphology. Moreover, large BC particles are more likely to exhibit loose fractal aggregates with thin coating, thus, is likely much different from core-shell structure. MAC in this case cannot be well depicted by using Mie model.*

Response: Thank you for your comments. In section 5.1 of our revised manuscript, the uncertainty caused by using idealized core-shell model was discussed by replacing the BC core with cluster-like aggregates calculated with multiple sphere T-matrix (MSTM) method. The relative deviation between MAC calculated by MSTM model and by core-shell Mie model was investigated. The results showed that when the size of BC core was smaller than 150 nm, the overall deviation was within 4 %, which indicated that Mie theory was a good approximation to the BC aggregates even when BC core reached 200 nm. When BC core was larger than 200 nm, MAC calculated by MSTM model increased with increasing thickness of shell. The deviations between MAC calculated by the idealized concentric core-shell model and letting BC particles be in the form of

cluster-like aggregates were overall within 15%.
* * *
**Specific comments:**

*1) Wavelength should be addressed when the absolute value of MAC is mentioned.*

Response: Thank you for your recommendation. Wavelength was addressed when the absolute value of MAC was mentioned.

*2) Line 13, what do the 'different core-shell structures' mean? Different core size and shell thickness?*

Response: Yes, 'different core-shell structures' meant different core sizes and shell thicknesses in this study. 'Different core-shell structures' was changed into 'different core sizes and shell thicknesses' in the revised manuscript to avoid ambiguity.

*3) Line 57–58, Bond and Bergstrom (2006) just suggested a consistent MAC for fresh (uncoated) BC particles.*

Response: This sentence was removed in the revised manuscript.

*4) Line 73, a more detailed but clear description of BCPMSD measurement should be addressed. From my understanding, major results and discussion presented in this study are based on the BCPMSD measurements (using DMA-AE51?) at Zhangqiu site. DMA-SP2 measurements at Taizhou, and comparisons of AE33 with PASS-3 at Taizhou and Beijing are mostly used to provide essential parameters (e.g., number fraction of BC-containing particles, multi-scattering correction factor for AE33, etc.) for the BCPMSD retrieval.*

Response: Thanks for your comments. More detailed description of BCPMSD measurement was addressed in section 2.2 in our revised manuscript.

Yes, the major results and discussion in this study were based on BCPMSD measurements using DMA-AE51 at Zhangqiu site. The SP2 measurements at Taizhou

as well as comparison between AE33 and PASS-3 were used to provide number fraction of BC-containing particles as well as multi-scattering correction factor for AE33.

*5) Line 112–115, the method to determine the size-resolved number fraction of BC-containing particles should be introduced briefly. How to deal with the effect of multicharged particles in the DMA-SP2 system? Why the number fraction of BC-containing particles at Taizhou can be used to represent that at Zhangqiu?*

Response: Thanks for your recommendation. The determination of the number fraction of BC-containing particle was introduced briefly in the text.

According to the study of Zhao et al. (2019), The peak height (H) of the aerosol scattering signal could be used to deal with multicharged particle. The probability distribution of H at a given selected mobility diameter had multiple modes, as Fig. 1 showed. The multiple modes corresponded to signals of multicharged particles and could be calculated with theory of DMA.

[Figure]

**Figure 1.** Figure S1 of the study by Zhao et al. (2019). The measured scattering signal distribution at diameter of 120 nm using ammonium sulfate.

Both Zhangqiu site (36°42'N, 117°30'E) and Taizhou site (32°35'N, 119°57'E) are in the east of China. They both experienced pollutions caused by industrialization and urbanization in the past several decades. Hence, the number fraction of BC-containing particle measured at Taizhou was representative and could be used as reference value for Zhangqiu.

**6) Line 120, why absorption coefficients measured by AE33 are 2.9 times those measured by PASS-3? Does this ratio mean the multi-scattering effect of the filter loading method? However, as mention in line 106, a compensation factor of 2.6 has been introduced to mitigate multiple scattering effect. Was the PASS-3 well calibrated before the measurement?**

Response: 2.9 was from the study by Zhao et al. (2020).

Yes, this ratio, namely the scattering correction factor, was used to correct multi-scattering effect.

In line 106, the factor of 2.6 was the scattering correction factor for AE51. And for AE33 was 2.9. We specified that 2.6 was for AE51.

**7) Line 147, although the mantle chemical species would not influence largely the results presented in this study, BC/OM mixtures are more likely existed in the atmosphere of studied regions.**

Response: Thanks for the comments. The wavelength used in this study was 880 nm. Previous study indicates aerosol absorption at 880 nm is mainly from BC (Ramachandran and Rajesh, 2007). Therefore, the influence of organic matter was neglected in this study.

**References**

Petzold, A., Ogren, J. A., Fiebig, M., Laj, P., Li, S. M., Baltensperger, U., Holzer-Popp, T., Kinne, S., Pappalardo, G., Sugimoto, N., Wehrli, C., Wiedensohler, A., and Zhang, X. Y.: Recommendations for reporting "black carbon" measurements, Atmospheric Chemistry and Physics, 13, 8365-8379, 10.5194/acp-13-8365-2013, 2013.

Ramachandran, S., and Rajesh, T. A.: Black carbon aerosol mass concentrations over Ahmedabad, an urban location in western India: Comparison with urban sites in Asia, Europe, Canada, and the United States, J. Geophys. Res.-Atmos., 112, 19, 10.1029/2006jd007488, 2007.

Zhao, G., Zhao, W. L., and Zhao, C. S.: Method to measure the size-resolved real part

of aerosol refractive index using differential mobility analyzer in tandem with single-particle soot photometer, Atmospheric Measurement Techniques, 12, 3541-3550, 10.5194/amt-12-3541-2019, 2019.

Zhao, G., Yu, Y., Tian, P., Li, J., Guo, S., and Zhao, C.: Evaluation and Correction of the Ambient Particle Spectral Light Absorption Measured Using a Filter-based Aethalometer, Aerosol and Air Quality Research, 20, 1833-1841, 10.4209/aaqr.2019.10.0500, 2020.

---

## Author Comment (AC2) · 16 Dec 2020

Response to Anonymous Referee #2

**Major comments:**

*1) The authors have ignored the recommendations as proposed by Petzold et al. (2013), recommendations that are generally accepted by the scientific community, on how black carbon (BC) should be reported when derived from instruments that measure light attenuation, i.e. filter based or photoacoustic sensor. BC derived from these techniques should be reported as equivalent BC, or eBC. If or when this paper is resubmitted, the title should reflect clearly that it is eBC that is being discussed, not BC.*

Response: Thanks for your recommendation. The term 'black carbon' (BC) was changed into 'equivalent BC'. As suggested by Petzold et al. (2013), equivalent BC was abbreviated to "EBC" in the revised manuscript.

*2) A large fraction of the introduction is devoted to the importance of BC for climate change due to radiative forcing. What the authors fail to understand is that in the context of their study, the corrections to the MAC that they are proposing is completely irrelevant. Sensors that measure light absorption like the Aethalometer, are already providing the necessary information that is relevant to climate change, i.e. it is not the mass concentration that is important it is the optical cross section. I will address this further below with respect to the mixing state of BC, but the primary point is that the mass concentration of BC is not important when doing radiative transfer calculations if you already have the primary measurements of the coefficients of scattering and absorption. The authors also mention that BC might be efficient CCN or IN, both true statements but again irrelevant with respect to their study. Hence, the introduction needs to be completely rewritten to explain the real relevance of the current study, and that is to set some error bounds on eBC derived from Aethalometer measurements and NOT a cutting edge, new methodology that will in any way improve the accuracy of such measurements.*

Response: Thank you for your comments. The introduction was rewritten to emphasize the importance of MAC correction when deriving EBC from light absorption based on filter-based instrument.

***3) This study should be written up as a detailed analysis of the uncertainties in the MAC related to the mixing state of BC, i.e. the refractive indices, real and imaginary, the wavelength of incident light, and the relative sizes of the core and shell. Secondly, in the introduction, it should be made quite clear how this analysis differs from the many others that have already been published.***

Response: Thank you for your comments. Detailed uncertainty analysis, including refractive indices, was input in section 5 of the revised manuscript. The discussions included uncertainties of MAC caused by using idealized core-shell model, using constant BC-containing particle fraction, and variation of RI. The influence of sizes of core and shell were discussed in the uncertainty analysis. With respect to wavelength, EBC is derived from $\sigma_{ab}$ at a specific wavelength, namely 880 nm. At 880 nm, aerosol absorption is mainly from BC (Ramachandran and Rajesh, 2007). At shorter wavelength, absorption of organic carbon is not negligible any more, leading to difficulty of extracting BC absorption from total absorption. Therefore, the wavelength dependency of MAC was not discussed since the main goal of this study was to derive EBC, and the organic component was not included in this study.

The variation of MAC due to mixing state was not considered when deriving EBC from $\sigma_{ab}$ in the previous studies, difference between this study and previous studies was input in the text. The motivation of this study was to propose a modified approach considering variation of MAC due to mixing state.

***4) The methodology that is discussed in this paper is being promoted as a way to derive a more accurate EBC but this is misleading because in order to apply this you need a lot of additional complementary information about the size distribution of the BC, the fraction of particles that are mixed with BC, etc. If you had all the necessary information to begin with, then you wouldn't even need to try and derive EBC using***

*a variable MAC because you would already have enough information to estimate BC without the light absorption instrument. This should be made quite clear in a resubmission of this paper.*

Response: Thank you and we agree with your comments. Filter based instruments such as AE33 are used in operational networks worldwide due to their advantages such as low cost, simplicity of operation, less maintenance and convenience for data processing. However, the EBC measured by AE33 is not accurate because it uses a constant MAC. The motivation of this study was to propose a method to consider the variation of MAC to make EBC measured by AE33 more accurate.

The size distribution of BC required in this study was size distribution of absorption measured by aethalometer, which could be achieve by DMA in tandem with AE51. As for the number fraction of particles mixed with BC ($N_{BC}$), it was a reference value in this study. Uncertainty analysis showed that derived EBC was not that sensitive to $N_{BC}$.

*5) It is my opinion that the modeling that is being discussed with this study has as much importance for setting the error bars on light absorption derived from the filter-based measurements as for setting error bars for deriving eBC. There are many corrections that have been proposed to adjust the light absorption measurements for the impact of overloading, filter matrix effects, etc., but perhaps the results from the current study could also be used to establish how mixed state BC leads to under/over estimates of the absorption coefficient. The authors should give this serious consideration if they want their study to have more relevancy than it does in its current state.*

Response: Thank you for your comments. As mentioned above, the filter-based instruments such as AE33 are widely used in operational networks due to their advantages. This study aimed to investigate the role of variation in MAC on the derived EBC by AE33. Besides correction to EBC, more discussions about the effect of mixing state on the absorption coefficient were input in the text.

**Specific comments:**

*1) Line 1, "determination of black carbon mass concentration from aerosol light absorption using variable mass absorption cross-section". Here and from here on out this is to be called "equivalent black carbon".*

Response: "Black carbon" was changed into "equivalent black carbon" in the text.

*2) Line 10, "the mass absorption cross-section (MAC) is a crucial parameter for converting light absorption coefficient ($\sigma_{ab}$) to mass equivalent BC concentration ($m_{BC}$)". Here and forward, change this into eBC.*

Response: $m_{BC}$ was modified as EBC in the revised manuscript here and forward.

*3) Line 11, "traditional filter-based instrument, such as AE33, uses a constant MAC of 7.77 $m^2$/g to derive $m_{BC}$, which may lead to uncertainty in $m_{BC}$." Add the wavelength that this is for.*

Response: Thanks for your recommendation. wavelength of 880 nm was appended to 7.77 $m^2$/g in the text.

*4) Line 22, "because of its highly absorbing properties in the visible spectral region, BC is considered to have a significant influence on global warming." By definition, "black" means all wavelengths, not just visible.*

Response: "In the visible spectral region" was deleted in the revised manuscript.

*5) Line25, "despite the importance of BC to climate, the global mean direct radiative forcing of BC particles still spans over a poorly constrained range of 0.2 – 1 $W/m^2$." Please clarify. I don't understand what this means.*

Response: This sentence was removed from the text to avoid ambiguity.

*6) Line 29, "to fully evaluate the influences of BC particles on solar radiation or*

*precipitation, more precise measurements of BC mass loading in the atmosphere are required." This is an incorrect argument for saying that more accurate measurements of BC are needed because instrument like the aethalometer measure light absorption directly without the need for converting it to eBC. With respect to the impact on clouds, what is needed is better measurements that can show just exactly how BC does form droplets or ice. Hence, an accurate MAC is not relevant for these impacts. The only impact that BC mass has that is important is on health or damage to building surfaces.*

Response: Thanks for your comments. This sentence was deleted in the revised manuscript to make the content more relevant to the correction to EBC.

*7) Line 31, "a variety of techniques have been developed to measure real-time BC mass concentrations." None of these measure BC mass concentrations.*

Response: These absorption measurement techniques was removed and instruments measuring BC mass concentration, such as SP2, OCEC, SP-AMS, was input in the text.

*8) Line 37 and line 38, "it measures real-time BC concentrations by converting the absorption coefficient ($\sigma_{ab}$) into mass equivalent BC concentrations ($m_{BC}$) through a constant mass absorption cross-section (MAC), which provides the BC absorption per unit mass." AE33 does not measure BC concentrations and the wavelength dependency of MAC has to be discussed at the very beginning.*

Response: "BC concentrations" was changed into "BC absorption" and "880 nm" was appended to MAC.

*8) Line 53, "a wide range of MAC (2 – 25 m²/g) has been reported in previous studies." This range is due to wavelength dependency. What is the range for a single frequency, especially for the one being used here?*

Response: Thanks for your comments. This sentence was changed into "A wide range of MAC has been reported in previous studies. For instance, Bond and Bergstrom (2006) reported MAC at 550 nm varying from 1.6 m²/g. Sharma et al. (2002) reported MAC

at 880 nm varying from 6.4 to 28.3 m$^2$/g." to make wavelength dependency clear.

**9) Line 62 to 64, "the hypothetical BC mixing state affects the corresponding absorption properties. It is critical to propose a method to infer m$_{BC}$ from light attenuation measurements considering aerosol size and the process by which BC aerosols mix with other aerosol components." Is this being proposed, completely independent of any other information about the environment?**

Response: The mixing state of BC was one of the important factors that affect the absorption properties of BC-containing particles. The size of aerosol was required to estimate the effect of mixing state on BC absorption. It was dependent on other information, such as refractive index (RI). The influence of RI on the uncertainty of MAC was discussed in the later content. This sentence was removed to avoid ambiguity.

**10) Line 70, "this modified method measures size-resolved m$_{BC}$ accurately and improves the evaluation of BC radiative forcing." How can a theoretical model "measure" eBC?**

Response: Thanks for your comment. "Measure" was modified into "estimate".

**11) Line 77, "the DMA (Differential Mobility Analyzer)-SP2 system measurements to determine the number fraction of BC-containing aerosols and to compare AE33 and the three-wavelength photoacoustic soot spectrometer (PASS-3) were conducted in Taizhou." What wavelength of AE33 are compared?**

Response: The wavelengths used for comparison between AE33 and PASS-3 were 405 nm, 532 nm and 781 nm. 405 nm, 532 nm and 781 nm are the wavelengths PASS-3 measures. The wavelengths AE33 measures are 370 nm, 470 nm, 520 nm, 590 nm, 660 nm, 880 nm and 950 nm. For AE33, 405 nm, 532 nm and 781 nm were calculated with wavelengths pairs of (370 nm, 470 nm), (520 nm, 590 nm) and (660 nm, 880 nm) through Ångström relationship:

$$\frac{\sigma_{ab}(\lambda_1)}{\sigma_{ab}(\lambda_2)} = \left(\frac{\lambda_1}{\lambda_2}\right)^{-\alpha_{ab}},$$

$$\sigma_{ab}(\lambda) = \sigma_{ab}(\lambda_1)\left(\frac{\lambda}{\lambda_1}\right)^{-\alpha_{ab}}.$$

Detailed description can be found in (Zhao et al., 2020). Wavelengths (405 nm, 532 nm and 781 nm) as well as the reference was appended to the manuscript.

*12) Line 84, "Meanwhile, from March 21, 2017 to April 9, 2017 at the Peking University site, the results from simultaneous measurements from AE51 (model 51, microAeth, USA) and AE33 were compared." What wavelength?*

Response: The wavelength of AE51 was 880 nm. Wavelength of "880 nm" was appended to "AE51 and AE33".

*13) Line 99, "the dry aerosol scattering coefficients at 525 nm were measured simultaneously by an integrated nephelometer (Ecotech 100 Pty Ltd., Aurora 3000) with a flow rate of 3 L/min." How does this wavelength correspond to the Aethalometer wavelengths?*

Response: The dry scattering coefficient at 525 nm here was used as a proxy of pollution level. At a specific wavelength, higher (lower) dry scattering coefficient could indicate a relatively polluted (clean) episode. Dry scattering coefficient at 525 nm was not used for comparison with light attenuation measured by aethalometer. "As an indicator of pollution level" was appended to the sentence.

*14) Line 105, "factor k was set as 0.004 and ATN is the measured light attenuation when particles load on the fiber filter of AE51." Where does this value come from?*

Response: "k = 0.004" was from the work by Zhao et al. (2019). "(Zhao et al., 2019b)" was appended to "0.004" in the manuscript.

*15) Line 114 − 115, "according to the measurements from Taizhou, only 17% of the ambient particles that contained BC averagely for bulk aerosol populations." This is*

***an incomplete sentence.***

Response: this sentence was modified into "according to the measurements from Taizhou, only 17% of the ambient particles contained BC averagely for bulk aerosol populations.".

***16) Line 116, "we adjusted the measured wavelengths of AE33 to the measured wavelengths of PASS-3 (405 nm, 532 nm, and 781 nm)." How the adjustment is made?***

Response: 405 nm, 532 nm and 781 nm are the wavelengths PASS-3 measures. The wavelengths AE33 measures are 370 nm, 470 nm, 520 nm, 590 nm, 660 nm, 880 nm and 950 nm. They are not consistent. For comparison, the wavelengths of AE33 were interpolated to the wavelengths of PASS-3 in this study. Specifically, For AE33, 405 nm, 532 nm and 781 nm were interpolated with wavelengths pairs of (370 nm, 470 nm), (520 nm, 590 nm) and (660 nm, 880 nm) through Ångström relationship:

$$\frac{\sigma_{ab}(\lambda_1)}{\sigma_{ab}(\lambda_2)} = \left(\frac{\lambda_1}{\lambda_2}\right)^{-\alpha_{ab}},$$

$$\sigma_{ab}(\lambda) = \sigma_{ab}(\lambda_1)\left(\frac{\lambda}{\lambda_1}\right)^{-\alpha_{ab}}.$$

More detailed description could be found in (Zhao et al., 2020). The interpolation method was added to the manuscript. "Adjusted" was changed into "interpolated".

***17) Line 182 – 183, "it should be pointed out that the retrieval algorithm of BCPMSD is based on the assumption that BC-containing particles of a fixed diameter are all core-shell mixed and the corresponding $D_{BC}$ for a specific $D_{particle}$ is same." A major assumption. Where is the sensitivity study that evaluates this assumption? This uncertainty analysis belongs in the main text, not in a supplement.***

Response: Thanks for your comments. The sensitivity study from the supplement was moved to the section 5.1 in the revised manuscript.

**Reference**

Petzold, A., Ogren, J. A., Fiebig, M., Laj, P., Li, S. M., Baltensperger, U., Holzer-Popp,

T., Kinne, S., Pappalardo, G., Sugimoto, N., Wehrli, C., Wiedensohler, A., and Zhang, X. Y.: Recommendations for reporting "black carbon" measurements, Atmospheric Chemistry and Physics, 13, 8365-8379, 10.5194/acp-13-8365-2013, 2013.

Ramachandran, S., and Rajesh, T. A.: Black carbon aerosol mass concentrations over Ahmedabad, an urban location in western India: Comparison with urban sites in Asia, Europe, Canada, and the United States, J. Geophys. Res.-Atmos., 112, 19, 10.1029/2006jd007488, 2007.

Zhao, G., Tao, J. C., Kuang, Y., Shen, C. Y., Yu, Y. L., and Zhao, C. S.: Role of black carbon mass size distribution in the direct aerosol radiative forcing, Atmospheric Chemistry and Physics, 19, 13175-13188, 10.5194/acp-19-13175-2019, 2019.

Zhao, G., Yu, Y., Tian, P., Li, J., Guo, S., and Zhao, C.: Evaluation and Correction of the Ambient Particle Spectral Light Absorption Measured Using a Filter-based Aethalometer, Aerosol and Air Quality Research, 20, 10.4209/aaqr.2019.10.0500, 2020.

---

## Author Comment (AC3) · 16 Dec 2020

Response to Anonymous Referee #3

**Major comments:**

*1) As the authors pointed, for the new BC, its shape is chain-like, not a spherical one, so how do you know this method is applicable for the measurement. How many parts of BC is newly generated and how many is old one is there a guess for that? Do you have some samples measured ASAP and others saved and wait some time to let them to be old one?*

Response: Thank you very much for your comments. We discussed the uncertainties caused by using idealized core-shell model in section 5 of our new manuscript. We replaced the spherical BC particle with cluster-like aggregates using multiple sphere T-matrix (MSTM) method. The results show that the deviations between the idealized concentric core-shell model and the cluster-like aggregates are overall within 15%. For BC core smaller than 200 nm, the deviations are within 4%. So, the method is applicable for the measurement.

After emitted into ambient environment, a pure BC particle will soon be coated. The absorption ability of the coated BC particle will be enhanced due to lensing effect. the absorption coefficient ($\sigma_{ab}$) of the coated BC particle will be larger that of pure BC particle. In our method, we do not limit the BC-containing particle that it has to be core-shell structure, it can also be a pure BC particle as long as the calculated $\sigma_{ab}$ matches measured $\sigma_{ab}$. So, we do not need to guess how many parts of BC is newly generated and how many parts of BC is old.

Sorry, we do not have sample measured ASAP and others saved and wait some time to let them to be old one. But according to the work of Peng et al. (2016), the aging time scale is ~ 4 hours.

**Specific comments:**

*1) Line 16, "with in" should be "within".*

Response: We changed "with in" into "within" in our new manuscript.

**2) Line 58, what's mean of "degree of MAC"?**

Response: "The degree of MAC" actually means "the value of MAC". We changed "the degree of MAC" into "the value of MAC" in our new manuscript to avoid ambiguity.

**3) "… Mie model incorporated with core-shell configuration hypothesis was applied in this study to assess the limitation of the constant …" should be simplified as "… Mie model with assumption of core-shell particles was …"**

Response: We changed "… Mie model incorporated with core-shell configuration hypothesis was …" into "… Mie model with assumption of core-shell particles was …" in our new manuscript.

**4) Line 68, "Based on the detailed…" The word "the" should be deleted.**

Response: we removed "the" in our new manuscript.

**5) Line 73, "The measured BC particle mass size distribution (BCPMSD) was obtained from the field campaign conducted at the Zhangqiu Meteorology Station (36°42'N, 117°30'E), Shandong Province. This field campaign lasted for about 1 month, from July 23, 2017 to August 24, 2017. The Zhangqiu observation site is located in the North China Plain (NCP) and is surrounded by farmland and residential areas, representing regional background conditions of the NCP." should be rewritten as "The BC particle mass size distribution (BCPMSD) was measured at Zhangqiu Meteorology Station (36°42'N, 117°30'E), Shandong Province, surrounded by farmland and residential areas and a typical site for regional background conditions of North China Plain (NCP). The field campaign lasted for about 1 month, from July 23, 2017 to August 24, 2017."**

Response: We changed this part into "The BC particle mass size distribution (BCPMSD) was measured at Zhangqiu Meteorology Station (36°42'N, 117°30'E), Shandong Province, surrounded by farmland and residential areas and a typical site for regional

background conditions of North China Plain (NCP). The field campaign lasted for about 1 month, from July 23, 2017 to August 24, 2017." in our new manuscript.

**6) Line 76, the last word "system" should be deleted.**

Response: We deleted "system" in our new manuscript.

**7) Line 77, "measurements to determine …" should be "is used to determine …".**

Response: We changed "measurements to determine …" into "is used to determine …" in our new manuscript.

**8) Line 78, "The suburban measurement site", the word "measurement" should be deleted.**

Response: We deleted the "measurement" in our new manuscript.

**9) Line 79, the word "the" before "Jianghuai Plain" should be deleted.**

Response: We deleted "the" before "Jianghuai Plain" in our new manuscript.

**10) Line 86 and 87, "All the measurements in the three sites were conducted in containers where ambient temperature was controlled within 24 ± 2 °C with a particle pre-impactor to remove particles larger than 10 μm from the input air stream." should be rewritten as "All the measurements in the three sites were conducted in temperature (24 ± 2 °C) controlled containers, and a particle pre-impactor is used to remove particles larger than 10 μm from the input airflow."**

Response: The sentence was changed into "All the measurements in the three sites were conducted in temperature (24 ± 2 °C) controlled containers, and a particle pre-impactor is used to remove particles larger than 10 μm from the input airflow." In our new manuscript.

**11) Line 92, "developed by (Ning et al., 2013). The instrument setup was further improved by Zhao et al. (2019b)." should be "developed by Ning et al. (2013) and**

*improved by Zhao et al. (2019b)".*

Response: we changed "developed by (Ning et al., 2013). The instrument setup was further improved by Zhao et al. (2019b)" into "developed by Ning et al. (2013) and improved by Zhao et al. (2019b)" in our new manuscript.

**12) Line 101, "that were used to represent air pollution conditions" should be deleted.**

Response: "that were used to represent air pollution conditions" was deleted in our new manuscript.

**13) Line 105, the variables of k and ATN should be italic.**

Response: k and ATN were changed into italic in our new manuscript.

**14) Line 108, "in this study" should be deleted.**

Response: "in this study" was deleted in our new manuscript.

**15) Beginning of line 115, word "from" should be "at" and the same for line 117.**

Response: "from" was changed into "at" in our new manuscript.

**16) Line 117 and 118, "with a measurement flowrate of" should be "with flowrate of".**

Response: "with a measurement flowrate of" was changed into "with flowrate of" in our new manuscript.

**17) Line 123, "… through a constant MAC value" should be "under assumption of a constant MAC".**

Response: "… through a constant MAC value" was changed into "under assumption of a constant MAC".

**18) Line 130, "an appropriate model simulation is needed for representing a single BC particle's optical properties." What's meaning of this sentence?**

Response: This sentence means that a proper model is required to simulate the optical parameters, such as the MAC, absorption coefficient, and scattering coefficient, of BC-containing particles to a good approximation. To avoid ambiguity, this sentence was changed into "a proper model is required to simulate the optical properties of BC-containing particles to a good approximation." in our new manuscript.

*19) Line 131, "There are three widely employed mixing states that are used to represent the structure of BC-containing aerosols" should be "Three widely employed mixing states are used to represent the structure of BC-carried aerosols".*

Response: The sentence was changed into "Three widely employed mixing states are used to represent the structure of BC-carried aerosols." in our new manuscript.

*20) Line 133, "… chain-like aggregates composed of small spheres" should be "chain-like aggregates of small spheres".*

Response: "chain-like aggregates composed of small spheres" was changed into "chain-like aggregates of small spheres" in our new manuscript.

*21) Line 139, "the spherical core and shell favor the Mie model" should be deleted.*

Response: "the spherical core and shell favor the Mie model" was deleted in our new manuscript.

*22) Line 140, "in this study" should be deleted.*

Response: "in this study" at line 140 was deleted in our new manuscript.

*23) Line 143, could you use other words for the section title?*

Response: The section title was changed to "Simulation of MAC for BC-containing particle using Mie theory".

*24) Line 147, the word "frequent" should be replace by "common".*

Response: the word "frequent" was replaced by "common" in our new manuscript.

**25) Line 150, "… at the wavelength of 880 nm, calculated using the Mie theory, has been presented" should be "… at wavelength of 880 nm are simulated with Mie scattering method."**

Response: "… at the wavelength of 880 nm, calculated using the Mie theory, has been presented" was changed into "… at wavelength of 880 nm are simulated with Mie scattering method." in our new manuscript.

**26) Line 151, "reported to vary with incident light wavelength" should be "dependent on light wavelength".**

Response: "reported to vary with incident light wavelength" was changed into "dependent on light wavelength" in our new manuscript.

**27) Line 152~153, "as BC particles can be emitted from different fuels and conditions, RI cannot be observed directly, with both real and imaginary part of RI varying over a significantly wide range" should be "due to different sources of BC, both the real and imaginary part of RI varies over a significantly wide range".**

Response: "as BC particles can be emitted from different fuels and conditions, RI cannot be observed directly, with both real and imaginary part of RI varying over a significantly wide range" was changed into "due to different sources of BC, both the real and imaginary part of RI varies over a significantly wide range" in our new manuscript.

**28) Line 157, "averaged values are illustrated …" Do you mean "mean values …"**

Response: Yes, "averaged values" are actually "mean values". To avoid ambiguity, "averaged values" was changed into "mean values" in our new manuscript.

**29) Please rewrite paragraph between line 168 and 173 to make it simple and clear.**

Response: The paragraph between line 168 and 173 was rewritten to make it simpler and clearer in our new manuscript.

***30) Line 174, the first sentence "The detailed iterative procedure is illustrated in Fig. 2." Should be reposition to the end of last paragraph, and the word "detailed" should be "deleted".***

Response: The first sentence at Line 174 was repositioned to the end of the paragraph and the word "detailed" was deleted in our new manuscript.

***31) Line 175, "represented" should be replace by "shown".***

Response: "represented" was replace by "shown" in our new manuscript.

***32) Line 175, "a simplified algorithm for deriving BCPMSD was proposed by considering Fig. 1 as a look-up table." Should be rewritten as "a simplified algorithm was proposed to derive BCPMSD through a pre-calculated look-up table."***

Response: "a simplified algorithm for deriving BCPMSD was proposed by considering Fig. 1 as a look-up table." was rewritten as "a simplified algorithm was proposed to derive BCPMSD through a pre-calculated look-up table." in our new manuscript.

***33) Line 195 and 196, words "finer mode" and "coarser mode" should be replaced by "fine mode" and "coarse mode", please read through the whole draft to replace other similar words.***

Response: "finer mode" and "coarser mode" was replaced by "fine mode" and "coarse mode" through the whole draft in our new manuscript.

***34) Line 198, "The results indicate that with the boundary of 280 nm, two opposite deviation tendencies exist." should be replaced by "the results show that there exist two opposite deviation trends before and after the turning point around 280nm."***

Response: "The results indicate that with the boundary of 280 nm, two opposite deviation tendencies exist." was replaced by "the results show that there exist two opposite deviation trends before and after the turning point around 280nm." in our new manuscript.

**35) Line 247, "The variations in on …" should be "The variation of …"**

Response: "The variations in on …" was changed into "The variation of …" in our new manuscript.

**36) Line 247, "all MACs in the look-up table in Fig. 1 are the mean values as the imaginary part and real part of BC RI varied over a wide range." What's the meaning of this sentence mean, please rewrite?**

Response: This sentence was rewritten as "for a MAC (880 nm) point at $(D_{particle}, D_{BC})$ of Fig. 1, it is actually a mean value averaged with respect to both real part of RI varied from 1.5 to 2.0 and imaginary part of RI varied from 0.5 to 1.1." in our new manuscript.

**37) Please rewrite the whole paragraph between line 247~260 to make it clear and simple.**

Response: the whole paragraph between line 247~260 was re written in our new manuscript to make it clear and simple.

**38) Line 454 to line 459, please rewrite caption for Figure 3 and make it easy to read. The same for the caption of Figure 4.**

Response: The captions for Fig. 3 and Fig. 4 were rewritten in our new manuscript to make it easy to read.

Peng, J. F., Hu, M., Guo, S., Du, Z. F., Zheng, J., Shang, D. J., Zamora, M. L., Zeng, L. M., Shao, M., Wu, Y. S., Zheng, J., Wang, Y., Glen, C. R., Collins, D. R., Molina, M. J., and Zhang, R. Y.: Markedly enhanced absorption and direct radiative forcing of black carbon under polluted urban environments, Proceedings of the National Academy of Sciences of the United States of America, 113, 4266-4271, 10.1073/pnas.1602310113, 2016.

---

## Author Response (AR1)

Response to Anonymous Referee #1

**Major comments:**

1) A new method by considering the variation in MAC is developed to obtain BC mass size distribution and then bulk BC mass concentration from size-resolved light absorption measurements. Size-resolved MAC calculated on the basis of core-shell Mie model is mainly discussed, which is determined by  $D_p$ -dependent  $D_{BC}$  and coating thickness. However, there are many assumptions in calculation processes, e.g., same DBC and coating thickness at each selected mobility size, a constant number fraction of BC-containing particles, etc. Meanwhile, measurements were not described clearly. Response: Thanks for your comments. The size resolved MAC in this study was based on core-shell Mie model. The influence of the BC aggregates on the MAC as well as the relative deviation between the core-shell model and BC aggregates were discussed in section 5.1 of the revised manuscript to evaluate the effects of the morphology on MAC. With respect to the assumptions used in this study, their uncertainties were discussed in section 5, such as the uncertainties caused by using idealized core-shell model (section 5.1), by using a constant BC-containing particle fraction (section 5.2) and by variation of refractive index (section 5.3). With respect to description of measurements, a more detailed description of our measurement was added in section 2.

2) The significance of this study should be also strengthened. In my point of view, compared to BC mass loading, the light absorption measurements are more required to evaluate the influences of BC particles on solar radiation. Thus, MAC is likely to be more important for converting bulk BC mass loading, which can be directly measured by using chemical method (e.g., Thermo Optical Reflection-EC) or laser-induced incandescence techniques (e.g., SP2-rBC), to light absorption in climate research. The current study is more important for obtaining BC mass size distribution from size-resolved absorption measurement. BC mass size distribution obtained from the DMA-AE51 measurement based on the new method is also suggested to compare

**with that obtained from the direct measurement from DMA-SP2 system, which has used in the field campaign.**

Response: Thanks and we agree with your comments. More sentences were added in this text to stress the significance. The main goal of this study was to derive equivalent BC mass concentration (EBC, after Petzold et al. (2013)) more precisely and obtain BC particle mass size distribution (BCPMSD) from size-resolved absorption measurement. MAC is an important variable that has to be discussed in the process. Derivation of the EBC and related uncertainties were more discussed to emphasize that our goal was to determine EBC more precisely.

3) The Mie model is likely to not suitable for the calculation of BC aggregates with large sizes. For a small BC particle (core), the mass equivalent diameter of the assumed BC sphere is much smaller than the wavelength (880 nm) resulting in a less effect of morphology to absorption. In this case, the Mie model is somewhat feasible for absorption estimation. However, for a large BC particle (core), its mass equivalent diameter is close to the wavelength (i.e., large size parameter); thus, the absorption is largely influenced by the morphology. Moreover, large BC particles are more likely to exhibit loose fractal aggregates with thin coating, thus, is likely much different from core-shell structure. MAC in this case cannot be well depicted by using Mie model.

Response: Thank you for your comments. In section 5.1 of our revised manuscript, the uncertainty caused by using idealized core-shell model was discussed by replacing the BC core with cluster-like aggregates calculated with multiple sphere T-matrix (MSTM) method. The relative deviation between MAC calculated by MSTM model and by core-shell Mie model was investigated. The results showed that when the size of BC core was smaller than 150 nm, the overall deviation was within 4 %, which indicated that Mie theory was a good approximation to the BC aggregates even when BC core reached 200 nm. When BC core was larger than 200 nm, MAC calculated by MSTM model increased with increasing thickness of shell. The deviations between MAC calculated by the idealized concentric core-shell model and letting BC particles be in the form of

cluster-like aggregates were overall within 15%.

**Specific comments:**

1) Wavelength should be addressed when the absolute value of MAC is mentioned. Response: Thank you for your recommendation. Wavelength was addressed when the absolute value of MAC was mentioned.

**2) Line 13, what do the 'different core-shell structures' mean? Different core size and shell thickness?**

Response: Yes, 'different core-shell structures' meant different core sizes and shell thicknesses in this study. 'Different core-shell structures' was changed into 'different core sizes and shell thicknesses' in the revised manuscript to avoid ambiguity.

**3) Line 57–58, Bond and Bergstrom (2006) just suggested a consistent MAC for fresh (uncoated) BC particles.**

Response: This sentence was removed in the revised manuscript.

4) Line 73, a more detailed but clear description of BCPMSD measurement should be addressed. From my understanding, major results and discussion presented in this study are based on the BCPMSD measurements (using DMA-AE51?) at Zhangqiu site. DMA-SP2 measurements at Taizhou, and comparisons of AE33 with PASS-3 at Taizhou and Beijing are mostly used to provide essential parameters (e.g., number fraction of BC-containing particles, multi-scattering correction factor for AE33, etc.) for the BCPMSD retrieval.

Response: Thanks for your comments. More detailed description of BCPMSD measurement was addressed in section 2.2 in our revised manuscript.

Yes, the major results and discussion in this study were based on BCPMSD measurements using DMA-AE51 at Zhangqiu site. The SP2 measurements at Taizhou

as well as comparison between AE33 and PASS-3 were used to provide number fraction of BC-containing particles as well as multi-scattering correction factor for AE33.

5) Line 112–115, the method to determine the size-resolved number fraction of BCcontaining particles should be introduced briefly. How to deal with the effect of multicharged particles in the DMA-SP2 system? Why the number fraction of BCcontaining particles at Taizhou can be used to represent that at Zhangqiu?

Response: Thanks for your recommendation. The determination of the number fraction of BC-containing particle was introduced briefly in the text.

According to the study of Zhao et al. (2019), The peak height (H) of the aerosol scattering signal could be used to deal with multicharged particle. The probability distribution of H at a given selected mobility diameter had multiple modes, as Fig. 1 showed. The multiple modes corresponded to signals of multicharged particles and could be calculated with theory of DMA.

**Figure 1.** Figure S1 of the study by Zhao et al. (2019). The measured scattering signal distribution at diameter of 120 nm using ammonium sulfate.

Both Zhangqiu site (36°42'N, 117°30'E) and Taizhou site (32°35'N, 119°57'E) are in the east of China. They both experienced pollutions caused by industrialization and urbanization in the past several decades. Hence, the number fraction of BC-containing particle measured at Taizhou was representative and could be used as reference value for Zhangqiu. 6) Line 120, why absorption coefficients measured by AE33 are 2.9 times those measured by PASS-3? Does this ratio mean the multi-scattering effect of the filter loading method? However, as mention in line 106, a compensation factor of 2.6 has been introduced to mitigate multiple scattering effect. Was the PASS-3 well calibrated before the measurement?

Response: 2.9 was from the study by Zhao et al. (2020).

Yes, this ratio, namely the scattering correction factor, was used to correct multiscattering effect.

In line 106, the factor of 2.6 was the scattering correction factor for AE51. And for AE33 was 2.9. We specified that 2.6 was for AE51.

**7) Line 147, although the mantle chemical species would not influence largely the results presented in this study, BC/OM mixtures are more likely existed in the atmosphere of studied regions.**

Response: Thanks for the comments. The wavelength used in this study was 880 nm. Previous study indicates aerosol absorption at 880 nm is mainly from BC (Ramachandran and Rajesh, 2007). Therefore, the influence of organic matter was neglected in this study.

**References**

Petzold, A., Ogren, J. A., Fiebig, M., Laj, P., Li, S. M., Baltensperger, U., Holzer-Popp,
T., Kinne, S., Pappalardo, G., Sugimoto, N., Wehrli, C., Wiedensohler, A., and Zhang,
X. Y.: Recommendations for reporting "black carbon" measurements, Atmospheric Chemistry and Physics, 13, 8365-8379, 10.5194/acp-13-8365-2013, 2013.

Ramachandran, S., and Rajesh, T. A.: Black carbon aerosol mass concentrations over Ahmedabad, an urban location in western India: Comparison with urban sites in Asia, Europe, Canada, and the United States, J. Geophys. Res.-Atmos., 112, 19, 10.1029/2006jd007488, 2007.

Zhao, G., Zhao, W. L., and Zhao, C. S.: Method to measure the size-resolved real part

of aerosol refractive index using differential mobility analyzer in tandem with singleparticle soot photometer, Atmospheric Measurement Techniques, 12, 3541-3550, 10.5194/amt-12-3541-2019, 2019.

Zhao, G., Yu, Y., Tian, P., Li, J., Guo, S., and Zhao, C.: Evaluation and Correction of the Ambient Particle Spectral Light Absorption Measured Using a Filter-based Aethalometer, Aerosol and Air Quality Research, 20, 1833-1841, 10.4209/aaqr.2019.10.0500, 2020. Response to Anonymous Referee #2

**Major comments:**

1) The authors have ignored the recommendations as proposed by Petzold et al. (2013), recommendations that are generally accepted by the scientific community, on how black carbon (BC) should be reported when derived from instruments that measure light attenuation, i.e. filter based or photoacoustic sensor. BC derived from these techniques should be reported as equivalent BC, or eBC. If or when this paper is resubmitted, the title should reflect clearly that it is eBC that is being discussed, not BC.

Response: Thanks for your recommendation. The term 'black carbon' (BC) was changed into 'equivalent BC'. As suggested by Petzold et al. (2013), equivalent BC was abbreviated to "EBC" in the revised manuscript.

2) A large fraction of the introduction is devoted to the importance of BC for climate change due to radiative forcing. What the authors fail to understand is that in the context of their study, the corrections to the MAC that they are proposing is completely irrelevant. Sensors that measure light absorption like the Aethalometer, are already providing the necessary information that is relevant to climate change, i.e. it is not the mass concentration that is important it is the optical cross section. I will address this further below with respect to the mixing state of BC, but the primary point is that the mass concentration of BC is not important when doing radiative transfer calculations if you already have the primary measurements of the coefficients of scattering and absorption. The authors also mention that BC might be efficient CCN or IN, both true statements but again irrelevant with respect to their study. Hence, the introduction needs to be completely rewritten to explain the real relevance of the current study, and that is to set some error bounds on eBC derived from Aethalometer measurements and NOT a cutting edge, new methodology that will in any way improve the accuracy of such measurements. Response: Thank you for your comments. The introduction was rewritten to emphasize the importance of MAC correction when deriving EBC from light absorption based on filter-based instrument.

3) This study should be written up as a detailed analysis of the uncertainties in the MAC related to the mixing state of BC, i.e. the refractive indices, real and imaginary, the wavelength of incident light, and the relative sizes of the core and shell. Secondly, in the introduction, it should be made quite clear how this analysis differs from the many others that have already been published.

Response: Thank you for your comments. Detailed uncertainty analysis, including refractive indices, was input in section 5 of the revised manuscript. The discussions included uncertainties of MAC caused by using idealized core-shell model, using constant BC-containing particle fraction, and variation of RI. The influence of sizes of core and shell were discussed in the uncertainty analysis. With respect to wavelength, EBC is derived from  $\sigma_{ab}$  at a specific wavelength, namely 880 nm. At 880 nm, aerosol absorption is mainly from BC (Ramachandran and Rajesh, 2007). At shorter wavelength, absorption of organic carbon is not negligible any more, leading to difficulty of extracting BC absorption from total absorption. Therefore, the wavelength dependency of MAC was not discussed since the main goal of this study was to derive EBC, and the organic component was not included in this study.

The variation of MAC due to mixing state was not considered when deriving EBC from  $\sigma_{ab}$  in the previous studies, difference between this study and previous studies was input in the text. The motivation of this study was to propose a modified approach considering variation of MAC due to mixing state.

4) The methodology that is discussed in this paper is being promoted as a way to derive a more accurate EBC but this is misleading because in order to apply this you need a lot of additional complementary information about the size distribution of the BC, the fraction of particles that are mixed with BC, etc. If you had all the necessary information to begin with, then you wouldn't even need to try and derive EBC using a variable MAC because you would already have enough information to estimate BC without the light absorption instrument. This should be made quite clear in a resubmission of this paper.

Response: Thank you and we agree with your comments. Filter based instruments such as AE33 are used in operational networks worldwide due to their advantages such as low cost, simplicity of operation, less maintenance and convenience for data processing. However, the EBC measured by AE33 is not accurate because it uses a constant MAC. The motivation of this study was to propose a method to consider the variation of MAC to make EBC measured by AE33 more accurate.

The size distribution of BC required in this study was size distribution of absorption measured by aethalometer, which could be achieve by DMA in tandem with AE51. As for the number fraction of particles mixed with BC ( $N_{BC}$ ), it was a reference value in this study. Uncertainty analysis showed that derived EBC was not that sensitive to  $N_{BC}$ .

5) It is my opinion that the modeling that is being discussed with this study has as much importance for setting the error bars on light absorption derived from the filterbased measurements as for setting error bars for deriving eBC. There are many corrections that have been proposed to adjust the light absorption measurements for the impact of overloading, filter matrix effects, etc., but perhaps the results from the current study could also be used to establish how mixed state BC leads to under/over estimates of the absorption coefficient. The authors should give this serious consideration if they want their study to have more relevancy than it does in its current state.

Response: Thank you for your comments. As mentioned above, the filter-based instruments such as AE33 are widely used in operational networks due to their advantages. This study aimed to investigate the role of variation in MAC on the derived EBC by AE33. Besides correction to EBC, more discussions about the effect of mixing state on the absorption coefficient were input in the text.

**Specific comments:**

1) Line 1, "determination of black carbon mass concentration from aerosol light absorption using variable mass absorption cross-section". Here and from here on out this is to be called "equivalent black carbon".

Response: "Black carbon" was changed into "equivalent black carbon" in the text.

2) Line 10, "the mass absorption cross-section (MAC) is a crucial parameter for converting light absorption coefficient ( $\sigma_{ab}$ ) to mass equivalent BC concentration ( $\underline{m}_{BC}$ )". Here and forward, change this into eBC.

Response: mBC was modified as EBC in the revised manuscript here and forward.

3) Line 11, "traditional filter-based instrument, such as AE33, uses a constant MAC of  $\frac{7.77 \text{ m}^2/\text{g}}{1000 \text{ m}^2/\text{g}}$  to derive  $m_{BC}$ , which may lead to uncertainty in  $m_{BC}$ ." Add the wavelength that this is for.

Response: Thanks for your recommendation. wavelength of 880 nm was appended to  $7.77 \text{ m}^2/\text{g}$  in the text.

4) Line 22, "because of its highly absorbing properties in the visible spectral region,
BC is considered to have a significant influence on global warming." By definition,
"black" means all wavelengths, not just visible.

Response: "In the visible spectral region" was deleted in the revised manuscript.

5) Line25, "despite the importance of BC to climate, the global mean direct radiative forcing of BC particles still spans over a poorly constrained range of  $0.2 - 1 W/m^2$ ." Please clarify. I don't understand what this means.

Response: This sentence was removed from the text to avoid ambiguity.

6) Line 29, "to fully evaluate the influences of BC particles on solar radiation or

precipitation, more precise measurements of BC mass loading in the atmosphere are required." This is an incorrect argument for saying that more accurate measurements of BC are needed because instrument like the aethalometer measure light absorption directly without the need for converting it to eBC. With respect to the impact on clouds, what is needed is better measurements that can show just exactly how BC does form droplets or ice. Hence, an accurate MAC is not relevant for these impacts. The only impact that BC mass has that is important is on health or damage to building surfaces.

Response: Thanks for your comments. This sentence was deleted in the revised manuscript to make the content more relevant to the correction to EBC.

**7) Line 31, "a variety of techniques have been developed to measure real-time BC mass concentrations." None of these measure BC mass concentrations.**

Response: These absorption measurement techniques was removed and instruments measuring BC mass concentration, such as SP2, OCEC, SP-AMS, was input in the text.

8) Line 37 and line 38, "it measures real-time BC concentrations by converting the absorption coefficient ( $\sigma_{ab}$ ) into mass equivalent BC concentrations ( $m_{BC}$ ) through a constant mass absorption cross-section (MAC), which provides the BC absorption per unit mass." AE33 does not measure BC concentrations and the wavelength dependency of MAC has to be discussed at the very beginning.

Response: "BC concentrations" was changed into "BC absorption" and "880 nm" was appended to MAC.

**8) Line 53, "a wide range of MAC $(2 - 25 \text{ m}^2/\text{g})$ has been reported in previous studies." This range is due to wavelength dependency. What is the range for a single frequency, especially for the one being used here?**

Response: Thanks for your comments. This sentence was changed into "A wide range of MAC has been reported in previous studies. For instance, Bond and Bergstrom (2006) reported MAC at 550 nm varying from 1.6 m2/g. Sharma et al. (2002) reported MAC

at 880 nm varying from 6.4 to 28.3  $m^2/g$ ." to make wavelength dependency clear.

9) Line 62 to 64, "the hypothetical BC mixing state affects the corresponding absorption properties. It is critical to propose a method to infer  $m_{BC}$  from light attenuation measurements considering aerosol size and the process by which BC aerosols mix with other aerosol components." Is this being proposed, completely independent of any other information about the environment?

Response: The mixing state of BC was one of the important factors that affect the absorption properties of BC-containing particles. The size of aerosol was required to estimate the effect of mixing state on BC absorption. It was dependent on other information, such as refractive index (RI). The influence of RI on the uncertainty of MAC was discussed in the later content. This sentence was removed to avoid ambiguity.

**10) Line 70, "this modified method measures size-resolved $m_{BC}$ accurately and improves the evaluation of BC radiative forcing." How can a theoretical model "measure" eBC?**

Response: Thanks for your comment. "Measure" was modified into "estimate".

11) Line 77, "the DMA (Differential Mobility Analyzer)-SP2 system measurements to determine the number fraction of BC-containing aerosols and to compare AE33 and the three-wavelength photoacoustic soot spectrometer (PASS-3) were conducted in Taizhou." What wavelength of AE33 are compared?

Response: The wavelengths used for comparison between AE33 and PASS-3 were 405 nm, 532 nm and 781 nm. 405 nm, 532 nm and 781 nm are the wavelengths PASS-3 measures. The wavelengths AE33 measures are 370 nm, 470 nm, 520 nm, 590 nm, 660 nm, 880 nm and 950 nm. For AE33, 405 nm, 532 nm and 781 nm were calculated with wavelengths pairs of (370 nm, 470 nm), (520 nm, 590 nm) and (660 nm, 880 nm) through Ångström relationship:

$$\frac{\sigma_{ab}(\lambda_1)}{\sigma_{ab}(\lambda_2)} = \left(\frac{\lambda_1}{\lambda_2}\right)^{-\alpha_{ab}},$$
$$\sigma_{ab}(\lambda) = \sigma_{ab}(\lambda_1) \left(\frac{\lambda}{\lambda_1}\right)^{-\alpha_{ab}}$$

Detailed description can be found in (Zhao et al., 2020). Wavelengths (405 nm, 532 nm and 781 nm) as well as the reference was appended to the manuscript.

**12) Line 84, "Meanwhile, from March 21, 2017 to April 9, 2017 at the Peking University site, the results from simultaneous measurements from AE51 (model 51, microAeth, USA) and AE33 were compared." What wavelength?**

Response: The wavelength of AE51 was 880 nm. Wavelength of "880 nm" was appended to "AE51 and AE33".

13) Line 99, "the dry aerosol scattering coefficients at 525 nm were measured simultaneously by an integrated nephelometer (Ecotech 100 Pty Ltd., Aurora 3000) with a flow rate of 3 L/min." How does this wavelength correspond to the Aethalometer wavelengths?

Response: The dry scattering coefficient at 525 nm here was used as a proxy of pollution level. At a specific wavelength, higher (lower) dry scattering coefficient could indicate a relatively polluted (clean) episode. Dry scattering coefficient at 525 nm was not used for comparison with light attenuation measured by aethalometer. "As an indicator of pollution level" was appended to the sentence.

14) Line 105, "factor k was set as 0.004 and ATN is the measured light attenuation when particles load on the fiber filter of AE51." Where does this value come from? Response: "k = 0.004" was from the work by Zhao et al. (2019). "(Zhao et al., 2019b)" was appended to "0.004" in the manuscript.

15) Line 114 – 115, "according to the measurements from Taizhou, only 17% of the ambient particles that contained BC averagely for bulk aerosol populations." This is

**an incomplete sentence.**

Response: this sentence was modified into "according to the measurements from Taizhou, only 17% of the ambient particles contained BC averagely for bulk aerosol populations.".

*16) Line 116, "we adjusted the measured wavelengths of AE33 to the measured wavelengths of PASS-3 (405 nm, 532 nm, and 781 nm)." How the adjustment is made?* Response: 405 nm, 532 nm and 781 nm are the wavelengths PASS-3 measures. The wavelengths AE33 measures are 370 nm, 470 nm, 520 nm, 590 nm, 660 nm, 880 nm and 950 nm. They are not consistent. For comparison, the wavelengths of AE33 were interpolated to the wavelengths of PASS-3 in this study. Specifically, For AE33, 405 nm, 532 nm and 781 nm were interpolated with wavelengths pairs of (370 nm, 470 nm), (520 nm, 590 nm) and (660 nm, 880 nm) through Ångström relationship:

$$\frac{\sigma_{ab}(\lambda_1)}{\sigma_{ab}(\lambda_2)} = \left(\frac{\lambda_1}{\lambda_2}\right)^{-\alpha_{ab}},$$
$$\sigma_{ab}(\lambda) = \sigma_{ab}(\lambda_1) \left(\frac{\lambda}{\lambda_1}\right)^{-\alpha_{ab}}$$

More detailed description could be found in (Zhao et al., 2020). The interpolation method was added to the manuscript. "Adjusted" was changed into "interpolated".

17) Line 182 - 183, "it should be pointed out that the retrieval algorithm of BCPMSD is based on the assumption that BC-containing particles of a fixed diameter are all core-shell mixed and the corresponding  $D_{BC}$  for a specific  $D_{particle}$  is same." A major assumption. Where is the sensitivity study that evaluates this assumption? This uncertainty analysis belongs in the main text, not in a supplement.

Response: Thanks for your comments. The sensitivity study from the supplement was moved to the section 5.1 in the revised manuscript.

| D / · /· | e   | • • •      |       | 1        |      | , ,•          | C      |
|-----------------|-----|------------|-------|----------|------|---------------|--------|
| Determination   | nt  | equivalent | hlack | carhon   | magg | concentration | trom•  |
|                 | UI. | cuuivaicht | DIACK | vai ovii | mass | Concent ation | IIVIII |

[revised manuscript text omitted]
 is introduced to determine  $m_{BC}$ -EBC from the measurement of thed  $\sigma_{ab}$ -at a given diameter. For At a given  $D_{particle}$  [=DBC + Tshell] selected by DMA, if DBC is prescribedassumed, the corresponding Tshell is determined is fixed. Combining the simultaneously measured PNSD particle number concentration (N(Dparticle)) by CPC downstream the DMA and the prescribed percentage of particles containing BCNBC, the number of BC-containing particles (NPC(Dparticle)) is then 带格式的: 下标

|---|----------------------|
|   |                      |

| 263 | determined at $D_{\text{particle}}$ . $\sigma_{ab}$ can then be calculated by Mie model with $D_{\text{particle}}$ , $D_{BC}$ and $N_{BC}(D_{\text{particle}})$ . Corresponding absorption |
|-----|--------------------------------------------------------------------------------------------------------------------------------------------------------------------------------------------|
| 264 | properties at the $D_{\text{particle}}$ with fixed $D_{BC}$ and $T_{\text{shell}}$ can be calculated using the Mie model. If the calculated $\sigma_{ab}$ matches measured                 |
| 265 | $\sigma_{ab}$ by AE51, then the prescribed $D_{BC}$ is considered as diameter of BC core at $D_{particle}$ . Else, $D_{BC}$ is changed until calculated $\sigma_{ab}$                      |
| 266 | equals measured $\sigma_{ab}$ . MAC can be calculated by Mie model with $D_{particle}$ , $D_{BC}$ and a presumed BC density. EBC at $D_{particle}$ is then                                 |
| 267 | derived by dividing measured $\sigma_{ab}$ by MAC. BCPMSD can then be derived through changing D particle selected by DMA. Hence, if                                            |
| 268 | the number concentration of BC-containing particles and $\sigma_{ab}$ at a given $D_{particle}$ are measured, we can infer the $D_{BC}$ -by closing the                                    |
| 269 | measured and the calculated $\sigma_{ob}$ . Then, the mac can be obtained from $D_{BC}$ for every $D_{particle}$ . Finally, the BCPMSD is derived.                                         |
| 270 | The detailed iterative procedure is illustrated in Fig. 2. As the absorption properties of BC particles in different coating states have                                                   |
| 271 | been evaluated with the Mie model, as represented shown in Fig. 1, a simplified algorithm was proposed for deriving BCPMSD                                                                 |
| 272 | was proposed by considering Fig. 1 as a through a pre-calculated look-up table. For every specificeach Dparticle selected by DMA, if                                                       |
| 273 | a $D_{BC}$ is assumed, the corresponding MAC of the particle can be derived from the look-up table. Then, the $\sigma_{ab}$ can be derived from                                            |
| 274 | the MAC, the assumed BC density (1.8 g/cm 3 in this study), and the number of BC-containing particles N BC (17% of the total number                                  |
| 275 | for every each $D_{particle}$ ). We adjusted the guessed $D_{BC}$ until the difference between calculated and measured $\sigma_{ab}$ was within an                                         |
| 276 | acceptable range (0.1%). Consequently, the $D_{BC}$ and thus the $\frac{m_{BC}}{EBC}$ at a given $D_{particle}$ was determined. The $\frac{m_{BC}}{EBC}$ at different                      |
| 277 | aerosol sizes were derived separately. Finally, the size-resolved $\frac{m_{BC}-EBC}{m_{BC}-EBC}$ and the bulk $\frac{m_{BC}-EBC}{m_{BC}-EBC}$ were obtained. The-detailed          |
| 278 | iterative procedure is illustrated in Fig. 2.                                                                                                                                              |

279 It should be pointed out that the retrieval algorithm of BCPMSD is based on the assumption that BC-containing particles of a fixed 280 diameter are all core-shell mixed and the corresponding DBC for a specific Dparticle is same. The uncertainties caused by idealized 281 core-shell model was discussed in section 5.1. Moreover, aA constant number percentage (17%) of BC-containing particles was 282 adopted in this study. However, the BC-containing particle fraction varied with the primary source, time, coagulation, and extent of 288 atmospheric process. The influence attributed to the constant fraction of BC-containing particles has been was discussed in section 284 2-of the supplement 5.2. Additionally, Bond et al. (2013) summarized the density for different graphitic materials. The density values are 1.8-2.1 g/cm3 for pure graphite, 1.8-1.9 g/cm3 for pressed pellets of BC, and 1.718 g/cm3 for fullerene soot. A constant density 285 286 (1.8 g/cm3) for BC was briefly used to calculate MAC and BC mass from the volume of particles with a diameter of DBC. Therefore, 28 the uncertainty of derived mBc-EBC in this study simply depends on the ratio of 1.8 g/cm3 and the real density. Finally, the MAC 288 values in the look-up table were the averaged mean values for different RI and the corresponding effects have beenwere discussed 289 in section 5.3.

**290 4 Results and discussion**

Figure 3 provides a comprehensive overview of the variations in measured and retrieved size-resolved parameters during the campaign. As evident from Fig. 3(a), for the BCPMSD derived by the new method, two modes were found, similar to the results of AE33. Figure 4(a) shows the averaged BCPMSD derived from the new method and AE33 during the campaign. The finerfine mode was located between 97 – 240 nm while the coarsercoarse mode was located between 240 – 602 nm. Figure 3(b) represents the relative deviations between the BCPMSD derived from the new proposed method and those derived from a constant MAC value of

|-----------------|
|                 |

296 7.77 m2/g at 880 nm. The results show that there exist two opposite deviation trends before and after the turning point around 29 280nm. The results indicate that with the boundary of 280 nm, two opposite deviation tendencies exist. For aerosol particles larger 298 than 280 nm, the mage EBC derived by the new method were mostly lower than those derived with the constant MAC value of 7.77 299 m2/g at 880 nm. In contrast, when aerosol particles were smaller than 280 nm, the mBe-EBC from the new method were significantly 300 higher than those calculated by the constant MAC, as shown in Fig. 3(c). Figure 3(c) shows the time series of size-resolved MAC 301 during the derivation process of BCPMSD. According to Fig. 3(c), for aerosol particles smaller than 280 nm, the corresponding 302 MAC was almost lower than 7.77 m2/g at 880 nm. This is because the MAC values of particles smaller than 280 nm are mostly 303 lower than 7.77 m2/g, as represented in Fig. 1. A smaller MAC implies a weaker absorption ability, which means that the same 304 measured  $\sigma_{ab}$  will correspond to an increased mBCEBC. Therefore, more BC mass loadingsEBC were derived from the new method. 305 For aerosol particles larger than 280 nm, in order to match the measured  $\sigma_{ab}$ , the corresponding  $D_{BC}$  were generally found to be in those regions of look-up table where the MAC values were larger than 7.77 m2/g at 880 nm\_(Fig. 3(c)). Thus, the BC mass loadings 306 30 for particles larger than 280 nm were found to be less than those calculated with the constant MAC value of 7.77 m2/g at 880 nm. 308 From Fig. 3(c), it can be seen that MAC varied from less than 4 m2/g to larger than 10 m2/g at 880 nm, which implies a large variability of the absorption ability of BC-containing particle. Therefore, if the conversion between  $m_{BC}$  and  $\sigma_{ab}$  is required, the 309 310 consideration of variation in mixing state is highly recommended. The simultaneously measured scattering coefficients at 525 nm 311 were introduced here to represent air pollution. As shown in Fig. 3(d), the observation station experienced different levels of 312 pollution. Deviations of mac EBC derived from the newly proposed method and the constant MAC at different aerosol sizes did not 313 show dependencies on pollution conditions.

314 Figure 3(e) shows the time series of mBC-EBC at finerfine and eoarsercoarse modes. Compared to the results of AE33, tThe mBC 315 EBC were more concentrated in the finerfine mode as compared to than in the coarser coarse mode. The mac-EBC at finerfine mode 316 were found to be higher than those at the coarsercoarse mode for 73% of the experiment campaign duration. The variation trends of bulk mBC-EBC calculated by considering the variations of MAC and a constant MAC were similar (Fig. 3(f)). The bulk mBC-EBC 31 31B calculated by the new method were higher than those derived by the constant MAC in 83% of the experiment campaign duration. 319 The mBC-EBC calculated from the new method and AE33 for different aerosol size ranges were statistically analyzed. As shown in 320 Fig. 4, for all mac-EBC of aerosols ranging between 97 - 602 nm and 97 - 280 nm derived from new method and AE33, strong 321 linear relationships were observed with correlation coefficients of 0.99 and 1.00, respectively. The ratios between the mBc-EBC 322 derived from AE33 and the new method for aerosol diameter ranges of 97 - 602 nm and 97 - 280 nm were 0.84 and 0.69, respectively, 328 indicating that the mBC-EBC obtained from AE33 was 16% lower for bulk aerosol particles and 31% lower for aerosols smaller than 32 280 nm. For the diameter range of 280 – 602 nm, MAC values varied significantly and the deviations in  $\frac{m_{PC}}{EBC}$  derived from the 325 new method and AE33 were divided into two types with a boundary of 0.7 ug/m3. If the mac-EBC derived from AE33 was lower 326 than 0.7  $\mu g/m^3$ , there was a relatively consistent ratio of 1.13 between the mpc-EBC from the new method and AE33, with a 327 correlation coefficient of 0.95. Therefore, BC mass loading from the AE33 algorithm was 13% higher for aerosol particles larger 328 than 280 nm and mac-EBC lower than 0.7 µg/m3. However, when the mac-EBC derived from AE33 was larger than 0.7 µg/m3, 
[revised manuscript text omitted]
 particle with small BC core. When Dparticle was fixed, the uncertainties decreased with increasing DBC. When DbC was 398 determined, the uncertainties did not change much with Tshell-, indicating the importance to quantify DBC for BC-containing particles 399 in order to reduce RI-related uncertainty in absorption. For pure BC particles, the uncertainties also decreased with increasing BC 400 particle size significantly from over 22% at 100 nm to less than 2% at 600 nm. Figure 57(b) shows the uncertainties when the 401 imaginary part was fixed at 0.8 and the real part ranged from 1.5 to 2.0 with an interval of 0.01. It can be seen that when the 402 imaginary part of RI was fixed, variations in the real part led to slight uncertainties. All the uncertainties were found to be below 408 14%. Figure  $\frac{57}{c}$  demonstrates the uncertainties when the real part was fixed at 1.75 and the imaginary part ranged from 0.5 to 1.1 404 with an interval of 0.01. Comparing Fig. 57(a) and 57(c), we can see that the patterns of MAC uncertainties were similar. Overall, 405 the uncertainties were dominated by the variations of the imaginary part and only slightly affected by variations in the real part. 406 Therefore, it is highly recommended to reduce the uncertainties in the imaginary part for a more precise absorption measurement. 40 The variations in of on mBC-EBC caused by the uncertainties in RI were further evaluated. As stated in section 3.2, allfor a MAC 408 (880 nm) point at MACs (Dparticle, DBC) of Fig. 1, it is a mean value averaged with respect to both real part of RI varied from 1.5 to 409 2.0 and imaginary part of RI varied from 0.5 to 1.1. in the look up table in Fig. 1 are the mean values as the imaginary part and real 410 part of BC RI varied over a wide range. Therefore, tThe mean MACs (880 nm) in the look up table plus corresponding standard 41 deviation (MAC + Std) and minus corresponding standard deviation (MAC - Std) were utilized used to show the uncertainties in mgc-EBC caused by variation of BC RI-of BC. As we can see from Fig. 68(a), irrespective of the MAC-values in look up table were 412 was MAC + Std or MAC - Std, there was no change in the mode of BCPMSD. The derived mac-EBC of all aerosols-particles 41B ranging from 97-602 nm increased when the MAC values used in the look up table were MAC - Std was used and decreased when 414 41 MAC + Std values were was used in the look up table. Compared to the bulk  $m_{BC}$ -EBC retrieved derived through the look up table withby mean MAC, those derived through the look up table withby MAC - Std were higher within 35% (Fig. 68(b)). The decrease 416 41 in the magnitude of derived magnitude derived was significantly less than the increase in the magnitude derived EBC caused by the MAC - Std values, Similarly, fFor aerosol particles at both finerfine and coarsercoarse modesmode particles, 418 419 the deviations in mPC-EBC caused by MAC + Std or MAC - Std were also within 35% (Fig. 6.8(c) and Fig. 6.8(d)). Meanwhile, the increase in the magnitude of derived  $\sigma_{ab}$  into mBC caused by the MAC-Std values was also significantly higher than the decrease in 420 the magnitude caused by the MAC + Std values. This sensitivity study indicated indicates that the accuracy of the retrieved derived 42 422 BCPMSD is sensitive to the accuracy of MAC-values in the look up table, especially when the real actual MACs are is less than the 42B mean MAC-values used in the look-up table.

An idealized concentric core shell model with a spherical BC core fully coated by sulfate was configured to study the MAC of BC
 aerosols and derive the mBC. However, freshly emitted BC particles were found to normally exist in the form of loose cluster-like

|-----------|-----------------|
| Y         |                 |

aggregates with numerous spherical primary monomers . Soon after, these aggregates become coated with other components and
 collapsed to a more compact form during the coating process . Therefore, the uncertainty in the idealized core shell configuration
 is discussed in section 3 of the supplement.

**429 6 Conclusions**

430 There was a significant variability in the MAC values of BC with the size of BC core and the thickness of coating, which exerted a 431 significant influence on the optical method for measuring  $m_{BC}$  deriving EBC. In this study, a new method was proposed to derive 432  $m_{BC}$  -EBC while considering the lensing effect of core-shell structure and subsequently the consequent MAC variations in MAC of 438 BC.

A look-up table describing the variations of MAC at 880 nm attributed to the coating state and size of BC core was established theoretically using Mie simulation and assuming a core-shell configuration for BC-containing aerosols. The MAC at 880 nm varied significantly with different sizes of core and shell from less than 2 m2/g to over 16 m2/g, indicating a large variation in absorption ability for BC-containing particle. Then, the mac-EBC at different aerosol sizes were derived by finding an appropriate BC core configured with a MAC value from the look-up table to close the calculated and measured  $\sigma_{ab}$ .

This newly proposed method was applied to a campaign measurement in the NCP. There were two modes for BCPMSD at the accumulation mode separated by 240 nm. For 73% of the cases, the  $m_{BC}$ -EBC of the finerfine mode were larger than those of the coarsercoarse mode during the measurement. The  $m_{BC}$ -
[revised manuscript text omitted]